# The mitochondrial iron transporter ABCB7 is required for B cell development, proliferation, and class switch recombination in mice

Michael Jonathan Lehrke, Michael Jeremy Shapiro, Matthew J Rajcula, Madeleine M Kennedy, Shaylene A McCue, Kay L Medina, Virginia Smith Shapiro*

Department of Immunology, Mayo Clinic, Rochester, United States

**Abstract** Iron-sulfur (Fe-S) clusters are cofactors essential for the activity of numerous enzymes including DNA polymerases, helicases, and glycosylases. They are synthesized in the mitochondria as Fe-S intermediates and are exported to the cytoplasm for maturation by the mitochondrial transporter ABCB7. Here, we demonstrate that ABCB7 is required for bone marrow B cell development, proliferation, and class switch recombination, but is dispensable for peripheral B cell homeostasis in mice. Conditional deletion of ABCB7 using Mb1-cre resulted in a severe block in bone marrow B cell development at the pro-B cell stage. The loss of ABCB7 did not alter expression of transcription factors required for B cell specification or commitment. While increased intracellular iron was observed in ABCB7-deficient pro-B cells, this did not lead to increased cellular or mitochondrial reactive oxygen species, ferroptosis, or apoptosis. Interestingly, loss of ABCB7 led to replication-induced DNA damage in pro-B cells, independent of VDJ recombination, and these cells had evidence of slowed DNA replication. Stimulated ABCB7-deficient splenic B cells from CD23-cre mice also had a striking loss of proliferation and a defect in class switching. Thus, ABCB7 is essential for early B cell development, proliferation, and class switch recombination.

**\*For correspondence:**
shapiro.virginia1@mayo.edu

**Competing interest:** The authors declare that no competing interests exist.

## Introduction

Iron is critical for numerous cellular processes including ATP production, cellular metabolism, and DNA replication and damage repair (*Fuss et al., 2015*; *Lane et al., 2015*; *Lill and Mühlenhoff, 2006*). However, labile iron (nonprotein bound) is highly toxic to cells as it can potently induce the formation of reactive oxygen species (ROS) through the Fenton reaction (*Lawen and Lane, 2013*). To protect organisms and cells from excess ROS, iron homeostasis is highly regulated at both a systemic and cellular level, which balances iron storage, transport, and utilization (reviewed by *Anderson and Frazer, 2017*; *Lawen and Lane, 2013*). It is thought that most cellular iron uptake in lymphocytes occurs through transferrin receptor 1 (TfR1; CD71), which binds transferrin-bound iron at the cell surface (*Lane et al., 2015*; *Lawen and Lane, 2013*; *Ned et al., 2003*). Intracellular iron can be stored in ferritin, which sequesters iron to prevent it from undergoing the Fenton reaction (*Lawen and Lane, 2013*). Alternatively, intracellular iron can be transported to the mitochondria for storage in mitochondrial ferritin or utilization in the biosynthesis of heme and iron-sulfur (Fe-S) clusters (*Lane et al., 2015*), making mitochondria essential for iron homeostasis. Typically, iron uptake and storage are controlled by intracellular iron levels through the activity of iron-responsive element-binding protein 1 (IRP1) and IRP2 (*Lane et al., 2015*; *Lawen and Lane, 2013*). However, many aspects of mitochondrial iron trafficking and storage are still not well understood (*Paul et al., 2017*; *Richardson et al., 2010*).

Fe-S clusters are important cofactors for numerous proteins involved in cellular metabolism, DNA replication, and DNA damage repair (*Fuss et al., 2015*; *Lane et al., 2015*; *Lill and Mühlenhoff, 2006*). These proteins include ferredoxin (*Lange et al., 2000*); components of NADH:ubiquinone oxidoreductase (Complex I) (*Ohnishi, 1998*); DNA primase (*Klinge et al., 2007*); all of the replicative DNA polymerases (*Netz et al., 2011*, reviewed by *Baranovskiy et al., 2018*; *Puig et al., 2017*); the helicases Dna2, FancJ, and XPD (*Mariotti et al., 2020*; *Rudolf et al., 2006*); and the glycosylases Endo III and MutY (*Cunningham et al., 1989*; *Porello et al., 1998*). Fe-S cluster intermediates are first biosynthesized in the mitochondria, after which they are transported to the cytoplasm for maturation (*Maio and Rouault, 2015*; *Rouault, 2012*). The inner mitochondrial membrane protein ABCB7 is important for Fe-S cluster maturation as it is thought to export an Fe-S-glutathione intermediate from the mitochondria to the cytoplasm (*Li and Cowan, 2015*; *Pondarré et al., 2006*; *Qi et al., 2014*; *Srinivasan et al., 2014*). Previous work demonstrated that ABCB7 is essential for life as ABCB7-deficient embryos failed to develop extra-embryonic tissues and tissue-specific deletion revealed a requirement for development and function of numerous cell types (*Pondarré et al., 2006*). ABCB7 was also found to be critical for hematopoiesis as conditional deletion with Mx1-cre resulted in rapid bone marrow failure with pancytopenia (*Pondarre et al., 2007*), but the requirement of ABCB7 for the development of specific hematopoietic lineages was not examined. HeLa cells have been shown to accumulate mitochondrial iron in the absence of ABCB7 (*Cavadini et al., 2007*); however, hepatocytes and endothelial cells appeared to be viable without ABCB7 and did not have iron accumulation (*Pondarré et al., 2006*). Therefore, some cell types appear to possess a compensatory mechanism to export Fe-S-glutathione intermediates in the absence of ABCB7, while ABCB7 is critical for this function in other cell types.

While iron levels have been linked to the proliferation and function of peripheral lymphocytes (*Jiang et al., 2019*; *Watanabe-Matsui et al., 2011*; *Yarosz et al., 2020* and reviewed by *Cronin et al., 2019*; *Kuvibidila et al., 2003*), their role in B cell development and peripheral B cell homeostasis has not been thoroughly characterized. Here, we show that ABCB7 is essential for B cell lymphopoiesis as B cell-specific conditional deletion of ABCB7 resulted in a severe block at the pro-B cell stage of development. We found that ABCB7-deficient pro-B cells accumulated iron, but did not have excess ROS or cell death. Gene expression changes indicated the near absence of pre-B cells. ABCB7-deficient pro-B cells also had reduced heavy chain recombination, and B cell development was restored upon introduction of a fully rearranged MD4 Hel-Ig transgenic B cell receptor (BCR). Interestingly, we found evidence that DNA damage was occurring in ABCB7-deficient pro-B cells independent of recombination. These results suggest that DNA damage was occurring during replication in the absence of ABCB7. Intriguingly, we found that ABCB7 was dispensable for peripheral B cell homeostasis. Using a B cell-specific CD23-cre to conditionally delete ABCB7 from peripheral B cells, we did not observe an obvious loss of B cell populations or numbers. However, we observed that CD23-cre ABCB7 cKO B cells had a striking defect in proliferation and class switching during in vitro class switch recombination (CSR) assays, but the severity of the defect was dependent upon stimulation signals received during culture. Pro-B cells bearing the fully rearranged MD4 Hel-Ig BCR had restored proliferation, likely as a result of IgM and IgD expression on these developing cells. These data demonstrate that ABCB7 is required for B cell development, proliferation, and CSR but is dispensable for peripheral B cell homeostasis.

## Results

### ABCB7 is required for pro-B cell development but is dispensable for peripheral B cell homeostasis

Previous literature demonstrated that ABCB7 is essential for hematopoiesis, but the role of ABCB7 in the development of specific hematopoietic lineages was not examined (*Pondarre et al., 2007*). To elucidate the role of ABCB7 in B cell development, ABCB7 was conditionally deleted by crossing ABCB7 floxed mice (*Clarke et al., 2006*) with Mb1-cre (*Hobeika et al., 2006*) transgenic mice, which express cre in early pre-pro-B cells (*Fahl et al., 2009*). Mb1-cre ABCB7 conditional knockout (cKO) mice had a severe reduction in B220+ CD19+ bone marrow B cells (*Figure 1A*, left-hand plots). A majority of the ABCB7-deficient B cells in Mb1-cre ABCB7 cKO mice were B220+ CD19+ CD43+ pro-B cells (*Figure 1A*, middle-left plots). Analysis of Hardy fractions (*Hardy et al., 1991*) in Mb1-cre ABCB7

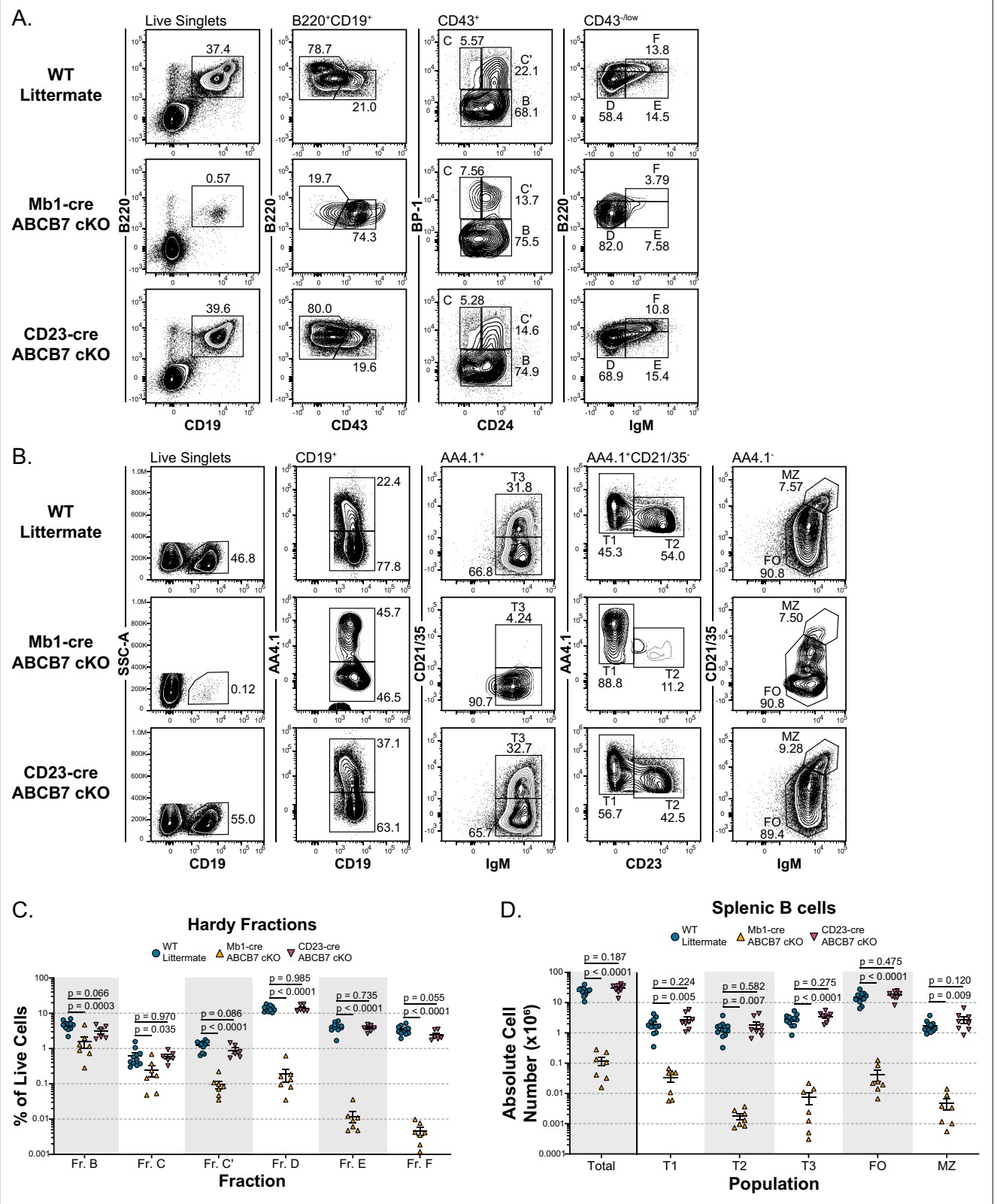

**Figure 1.** ABCB7 is required for pro-B cell development but not peripheral B cell homeostasis. (**A**) Flow cytometry analysis of B cell development in bone marrow from wild-type (WT), Mb1-cre ABCB7 conditional knockout (cKO), and CD23-cre ABCB7 cKO mice. Pro-B cells were divided into Hardy fractions as follows: Fr. B (B220$^+$ CD19$^+$ CD43$^+$ BP-1$^-$), Fr. C (B220$^+$ CD19$^+$ CD43$^+$ CD24$^{lo}$ BP-1$^+$), and Fr. C' (B220$^+$ CD19$^+$ CD43$^+$ CD24$^{hi}$ BP-1$^+$), Fr. D (B220$^+$ CD19$^+$ CD43$^{-/low}$ sIgM$^-$), Fr. E (B220$^+$ CD19$^+$ CD43$^{-/low}$ sIgM$^+$), and Fr. F (B220$^{hi}$ CD19$^+$ CD43$^{-/low}$ sIgM$^+$). Contour plots are representative of six independent

*Figure 1 continued on next page*

*Figure 1 continued*

experiments (total of 6–11 mice/group). (**B**) Flow cytometry analysis of splenic B cell populations in WT, Mb1-cre ABCB7 cKO, and CD23-cre ABCB7 cKO mice. Populations were identified by gating on CD19+ splenocytes: transitional type 1 (T1; AA4.1+CD21/35- IgM+ CD23-), transitional type 2 (T2; AA4.1+ CD21/35- IgM+ CD23+), transitional type 3 (T3; AA4.1+CD21/35+IgM+), follicular (FO; AA4.1- CD21/35+ IgM+), and marginal zone (MZ; AA4.1- CD21/35hi IgMhi). Contour plots are representative of seven independent experiments (total of 7–12 mice/group). (**C**) Graph showing the percentage of total live bone marrow cells for each Hardy fraction in (**A**). (**D**) Graph showing absolute cell numbers of splenic B cell populations in (**B**). (**C, D**) Lines represent the mean ± SEM. Statistics were obtained by using a one-way ANOVA with Dunnett's test for multiple comparisons.

The online version of this article includes the following figure supplement(s) for figure 1:

**Figure supplement 1.** Analysis of pro-B cell block in Mb1-cre ABCB7 conditional knockout (cKO) mice.

**Figure supplement 2.** ABCB7 deletion efficiency in CD23-cre ABCB7 conditional knockout (cKO) peripheral B cells.

cKO mice revealed a significant decrease in the proportion of developing B cells starting at fraction (Fr.) B, which had a threefold reduction when quantified as a percentage of live bone marrow cells (*Figure 1C*). There was also a nearly threefold reduction in the proportion of Fr. C cells in these mice (*Figure 1C*). A 13-fold decrease in the proportion of Fr. C' cells and a nearly 70-fold decrease in the proportion of Fr. D cells in Mb1-cre ABCB7 cKO mice indicated a severe block during pro-B development (*Figure 1A*, middle-right plots, *Figure 1C*). Quantification of the absolute cell numbers for each Hardy fraction confirmed a striking block during pro-B development and loss of later B cell development stages in Mb1-cre ABCB7 cKO mice (*Figure 1—figure supplement 1A*). The ratio of the number of Fr. C cells over the number of Fr. C' cells also supported a pro-B cell development block as Mb1-cre ABCB7 cKO mice had more Fr. C (pro-B) cells than Fr. C' (large pre-B) cells (*Figure 1—figure supplement 1B*). Very few IgM+ cells were observed in Mb1-cre ABCB7 cKO mice, with dramatic reductions in the proportions and numbers of naïve Fr. E cells and recirculating Fr. F cells (*Figure 1A*, right-hand plots, *Figure 1C*, *Figure 1—figure supplement 1A*). These data indicate ABCB7-deficient cells were blocked at the pro-B cell stage and failed to continue development into pre-B cells. Because of this, there was a significant decrease in the number of peripheral CD19+ B cells in the spleen of Mb1-cre ABCB7 cKO mice (*Figure 1B*, left-hand plots, *Figure 1D*).

To determine if ABCB7 was required for establishment or maintenance of peripheral B cells, ABCB7 was conditionally deleted by crossing ABCB7 floxed mice with CD23-cre transgenic mice (*Kwon et al., 2008*), in which cre expression is under the control of the B cell-specific *Cd23* promoter that is induced during the progression from transitional T1 to T2 B cell development (*Kondo et al., 1994*). These mice also express a human CD5 (huCD5) reporter linked to cre expression via an IRES (*Kwon et al., 2008*). CD23-cre ABCB7 cKO mice had normal proportions of each Hardy fraction in the bone marrow (*Figure 1A and C*), although the number of Fr. B and Fr. C' cells was slightly reduced in these mice (*Figure 1—figure supplement 1A*). Expression of the huCD5 reporter was only observed in mature, recirculating cells (Fr. F) and was absent from any pro- or pre-B cell (Fr. B-D; *Figure 1—figure supplement 2A*), which was expected as CD23-cre is expressed in the periphery and Fr. F cells are recirculating. No differences were observed in the proportion or absolute number of CD19+ B cells in the spleen of CD23-cre ABCB7 cKO mice (*Figure 1B*, left-hand plots, *Figure 1D*), further suggesting that bone marrow B cell development is normal in these mice. There were no differences observed in the numbers of T1, T2, T3, follicular (FO), or marginal zone (MZ) B cells in CD23-cre ABCB7 cKO mice (*Figure 1B and D*), implying that peripheral B cell homeostasis in these mice was also unaffected by the absence of ABCB7. Expression of the huCD5 reporter was largely absent on the majority of T1 cells, while expressed on T2, a large majority of T3, FO, and MZ B cells (*Figure 1—figure supplement 2B*), confirming that the B cell-specific *Cd23* promoter turns on at the transition from T1 to T2 B cells. Thus, using CD23-cre, the role of ABCB7 at the T1 stage cannot be analyzed. Additionally, quantitative PCR (qPCR) analysis confirmed the deletion of ABCB7 in sorted FO and MZ B cells from CD23-cre ABCB7 cKO mice (*Figure 1—figure supplement 2C*). These data demonstrate that ABCB7 is required for B cell development in the bone marrow, particularly in pro-B cells, but is dispensable for peripheral B cell homeostasis in the spleen.

## Gene expression changes confirm absence of pre-B cells in Mb1-cre ABCB7 cKO mice

B cell development is dependent on the concerted activity of several critical transcription factors that activate the early B cell developmental program, inducing B cell specification and commitment, including Early B-Cell Factor 1 (EBF1) (*Medina et al., 2004*; *O'Riordan and Grosschedl, 1999*), E2A (E47; *Tcf3*) (*Bain et al., 1994*; *Kwon et al., 2008*; *O'Riordan and Grosschedl, 1999*), Forkhead Box O1 (FOXO1) (*Dengler et al., 2008*), and PAX5 (*Fuxa et al., 2004*; *Nutt et al., 1997*; *Souabni et al., 2002*). Additionally, combined heterozygous loss of EBF1 and E2A, PAX5, and/or FOXO1 can also disrupt B cell commitment and development (*Lin et al., 2010*; *Ungerbäck et al., 2015*). Therefore, protein expression of EBF1, E47, FOXO1, and PAX5 in ABCB7-deficient pro-B cells was analyzed by flow cytometry. No significant decrease in expression of any of these critical transcription factors was observed in Fr. C cells (B220$^+$ CD19$^+$ CD43$^+$ BP-1$^+$) in Mb1-cre ABCB7 cKO mice (*Figure 2A–D*). EBF1 expression had a slight, but significant, increase in expression in ABCB7-deficient Fr. C cells (*Figure 2A*). These results suggest that the B cell transcriptional program is intact in ABCB7-deficient pro-B cells.

As cells progress from the pro- to pre-B cell stage, signaling from the pre-B cell receptor (pre-BCR) induces upregulation of several transcription factors and surface markers, including IKAROS (*Ikzf1*) (*Ferreirós-Vidal et al., 2013*), AIOLOS (*Ikzf3*) (*Thompson et al., 2007*), interferon regulatory factor 4 (IRF4) (*Lu, 2003*), CD2, and CD25 (IL-2Rα) (*Rolink et al., 1994*; *Yagita et al., 1989*). Flow cytometric analysis of IKAROS, AIOLOS, and IRF4 revealed that developing B cells in Mb1-cre ABCB7 cKO mice failed to upregulate these transcription factors (*Figure 2E–G*). Additionally, the proportions of developing B cells with surface expression of CD2 and CD25, hallmarks of transition to the pre-B cell stage, were markedly decreased in Mb1-cre ABCB7 cKO mice (*Figure 2H and I*). Upon successful rearrangement of the immunoglobulin μ heavy chain (μHC), signals from the pre-BCR induce downregulation of recombination machinery components at the transcriptional and protein level: terminal deoxynucleotidyl transferase (TdT), Rag1, and Rag2 (*Galler et al., 2004*; *Grawunder et al., 1995*; *Li et al., 1996*). TdT expression was examined via flow cytometry and was found to be maintained in ABCB7-deficient Fr. C cells (*Figure 2J*, right plot). Interestingly, TdT was also highly expressed in Fr. B cells in Mb1-cre ABCB7 cKO mice as compared to WT mice (*Figure 2J*, left plot). There was not a statistically significant difference in Rag1 and Rag2 transcripts in sorted Fr. B and Fr. C cells (*Figure 2K*). Additionally, bone marrow from Mb1-cre ABCB7 cKO mice failed to yield pre-B cell colonies after 8 days in an IL-7-dependent colony-forming unit (CFU-pre-B) assay (*Figure 2—figure supplement 1*), confirming the absence of pre-B cells. Collectively, these experimental findings show that transcription factors required for B cell specification and commitment are normally expressed in ABCB7-deficient B cells, but changes in gene expression concomitant with successful traversal of the pro- to pre-B cell transition are altered.

## ABCB7-deficient pro-B cells have increased intracellular iron, but lack evidence of iron-related cellular stress

Excessive labile iron (nonprotein bound) and heme can potently induce the generation of ROS or lead to ferroptosis, an iron-dependent form of cell death (*Lawen and Lane, 2013*; *Li et al., 2020*). Because it has been demonstrated that HeLa cells accumulate mitochondrial iron in the absence of ABCB7 (*Cavadini et al., 2007*), iron levels and the mitochondria were examined in ABCB7-deficient pro-B cells. We used Phen Green SK diacetate (Phen Green), which emits green fluorescence that is quenched in the presence of heavy metal ions such as iron (*Petrat et al., 1999*), to quantify intracellular iron levels. Mb1-cre ABCB7 cKO pro-B cells (B220$^+$ CD19$^+$ CD43$^+$) had a fourfold increase in the proportion of cells with quenched Phen Green fluorescence compared to WT (*Figure 3A*), consistent with elevated cellular iron levels. A compensatory mechanism for export of Fe-S-glutathione intermediates in the absence of ABCB7 has been hypothesized as ABCB7-deficient liver and endothelial cells are viable and do not have increased iron accumulation (*Pondarré et al., 2006*). Intriguingly, peripheral CD19$^+$ cells from CD23-cre ABCB7 cKO mice did not have iron accumulation as detected by Phen Green quenching (*Figure 3—figure supplement 1*), indicating that they possess a compensatory export mechanism for removing mitochondrial Fe-S-glutathione intermediates in the absence of ABCB7.

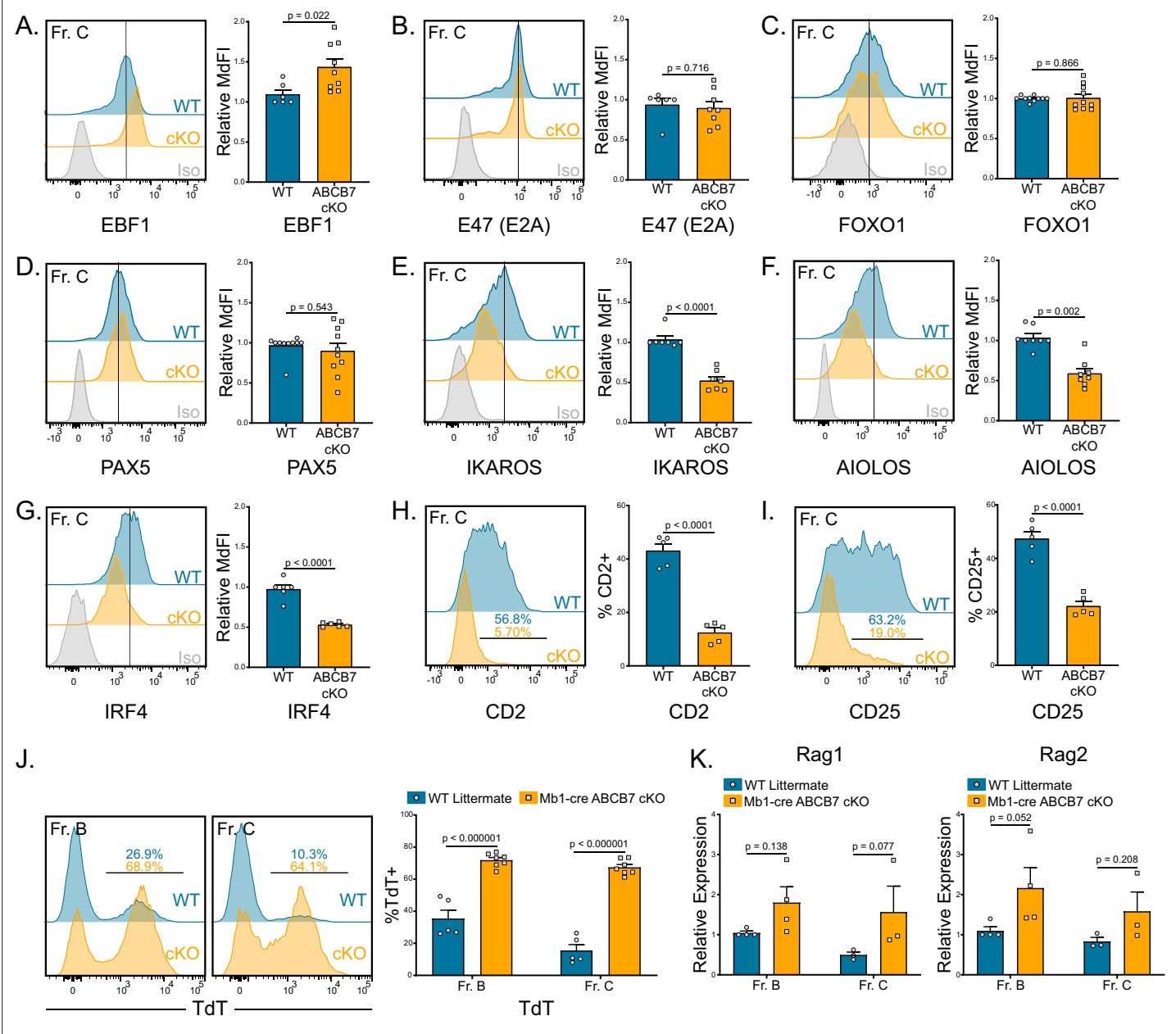

**Figure 2.** Gene expression changes confirm absence of pre-B cells in Mb1-cre ABCB7 conditional knockout (cKO) mice. Analysis of critical transcription factors in wild-type (WT) and Mb1-cre ABCB7 cKO Fr. C cells (B220⁺CD19⁺CD43⁺BP-1⁺). (**A–G**) Intracellular flow cytometry analysis of EBF1 (**A**), E47 (E2A) (**B**), FOXO1 (**C**), PAX5 (**D**), IKAROS (**E**), AIOLOS (**F**), and IRF4 (**G**) expression. Quantification of MdFI is shown on the right of each plot. Isotype controls are shown in gray. Offset histograms are representative of at least three independent experiments (total of 6–10 mice/group). (**H, I**) Flow cytometry analysis of CD2 (**H**) and CD25 (**I**) expression. Indicated values are the proportion of Fr. C cells positive for either marker, and quantifications are shown on the right of each plot. Offset histograms are representative of three independent experiments (total of five mice/group). (**J**) Intracellular flow cytometry analysis of TdT expression in Fr. B and Fr. C cells. Indicated values are the proportion of cells positive for TdT expression, and quantifications are shown on the right. Offset histograms are representative of three independent experiments (total of 5–7 mice/group). (**K**) Quantitative real-time PCR analysis of Rag1 and Rag2 expression in sorted Fr. B and Fr. C cells. 18S rRNA was used as an endogenous control, and relative expression values were normalized to expression in WT Fr. B cells. Results were obtained from three independent experiments (total of 3–4 mice/group). (**A–K**) Error bars represent SEM, and p-values are indicated above the data. Statistics were obtained by using an unpaired Student's *t*-test.

The online version of this article includes the following figure supplement(s) for figure 2:

**Figure supplement 1.** Failure to generate pre-B cell colony-forming unit (CFU) from Mb1-cre ABCB7 conditional knockout (cKO) mice.

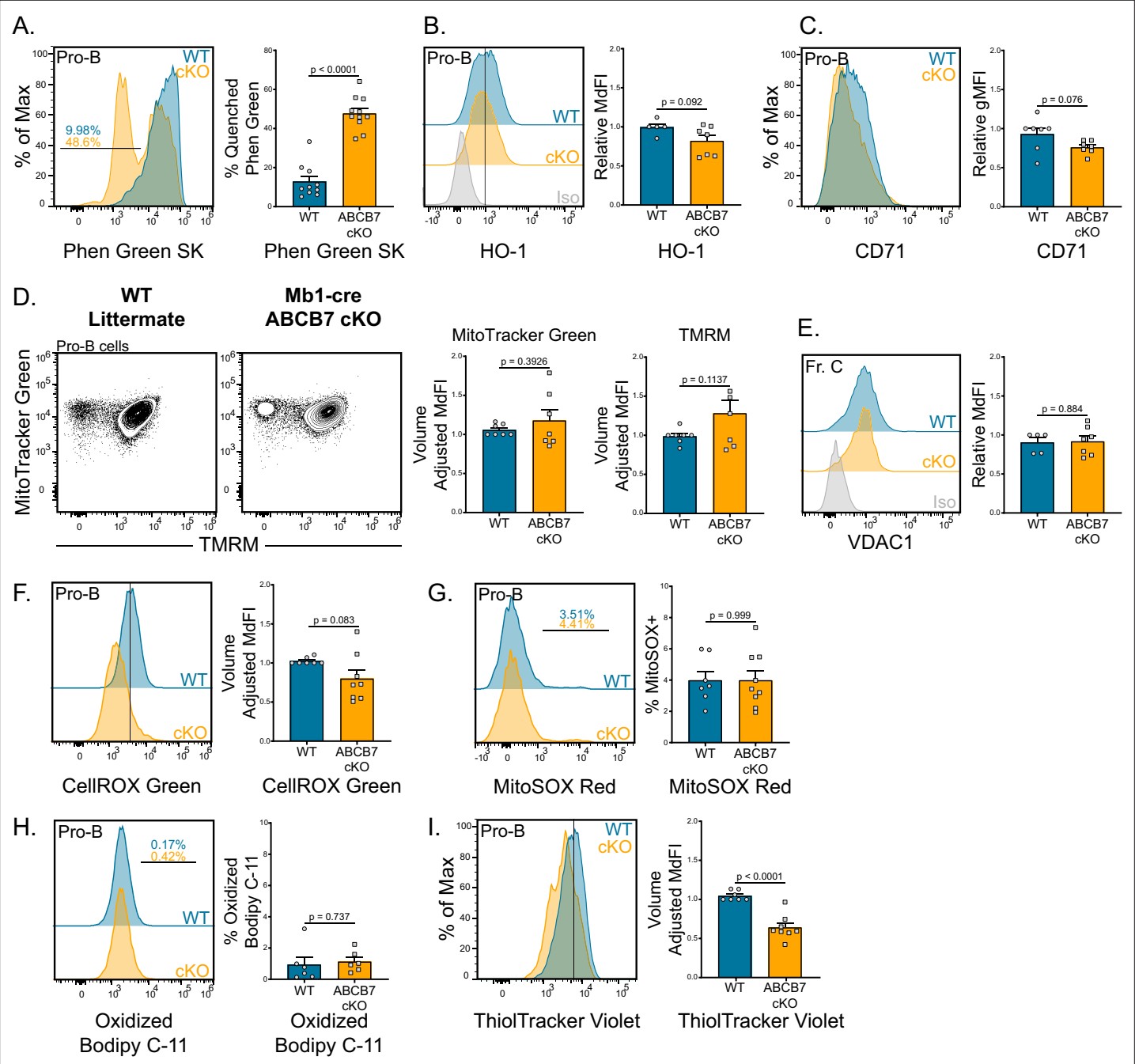

**Figure 3.** Iron accumulation in ABCB7-deficient pro-B cells. Analysis of mitochondria, iron accumulation, and reactive oxygen species (ROS) in wild-type (WT) and Mb1-cre ABCB7 conditional knockout (cKO) pro-B cells (B220+ CD19+ CD43+). (**A**) Flow cytometry analysis of Phen Green SK fluorescence quenching by heavy metal atoms. Indicated values are the proportion of cells with quenched fluorescence, and quantification is shown on the right. Overlaid histogram is representative of five independent experiments (total of 10–11 mice/group). (**B**) Intracellular flow cytometry analysis of HO-1 expression. Quantification of HO-1 MdFI is shown on the right. Offset histogram is representative of three independent experiments (total of 5–7 mice/group). (**C**) Flow cytometry analysis of CD71 expression. Quantification of CD71 gMFI is shown on the right. Overlaid histogram is representative of three independent experiments (total of seven mice/group). (**D**) Flow cytometry analysis of mitochondria abundance (MitoTracker Green) and membrane potential (tetramethylrhodamine methyl ester [TMRM]). Quantification of MitoTracker Green volume-adjusted MdFI and TMRM volume-adjusted MdFI is shown on the right. Contour plots are representative of four independent experiments (total of seven mice/group). (**E**) Intracellular flow cytometry analysis of VDAC1 expression in Fr. C cells (B220+ CD19+ CD43+ BP-1+). Quantification of VDAC1 MdFI is shown on the right. Offset histogram is representative of three independent experiments (total of 5–7 mice/group). (**F**) Flow cytometry analysis of CellROX Green ROS detection probe. Quantification of CellROX Green volume-adjusted MdFI is shown on the right. Offset histogram is representative of five independent experiments

*Figure 3 continued on next page*

Figure 3 continued

(total of 7–8 mice/group). (**G**) Flow cytometry analysis of MitoSOX Red mitochondrial ROS detection probe. Indicated values are the proportion of cells positive for MitoSOX Red dye, and quantification is shown on the right. Offset histogram is representative of four independent experiments (total of 7–9 mice/group). (**H**) Flow cytometry analysis of Bodipy C-11 lipid peroxidation probe. Indicated values are the proportion of cells positive for oxidized Bodipy C-11, and quantification is shown on the right. Offset histogram is representative of three independent experiments (total of six mice/group). (**I**) Flow cytometry analysis of ThiolTracker Violet glutathione detection agent. Quantification of ThiolTracker Violet volume-adjusted MdFI is shown on the right. Overlaid histogram is representative of four independent experiments (total of 7–8 mice/group). (**A–H**) Error bars represent SEM, and p-values are indicated above the data. Statistics were obtained by using an unpaired Student's *t*-test.

The online version of this article includes the following figure supplement(s) for figure 3:

**Figure supplement 1.** Splenic B cells in CD23-cre ABCB7 conditional knockout (cKO) do not have iron accumulation.

In addition to Fe-S clusters, iron-containing heme is synthesized in the mitochondria and has been shown to regulate differentiation and class switching in peripheral B cells (*Lane et al., 2015*; *Lawen and Lane, 2013*). Because heme synthesis occurs in the mitochondria, mitochondrial iron could be shunted into the heme synthesis pathway for export in the absence of ABCB7. The expression of heme oxygenase-1 (HO-1) was examined as a surrogate marker for heme levels as HO-1 expression is upregulated when intracellular heme levels increase (*Watanabe-Matsui et al., 2011*). Expression of HO-1 in Mb1-cre ABCB7 cKO pro-B cells was not significantly different compared to WT pro-B cells (*Figure 3B*). Therefore, there does not appear to be an increase in intracellular heme in ABCB7-deficient pro-B cells. ABCB7-deficient HeLa cell cultures displayed a cytoplasmic iron starvation phenotype, characterized by upregulated CD71 expression, despite accumulating mitochondrial iron (*Cavadini et al., 2007*). CD71 expression was not increased in ABCB7-deficient pro-B cells (*Figure 3C*), suggesting that they did not possess a similar cytoplasmic iron starvation phenotype. To assess mitochondria in ABCB7-deficient pro-B cells, MitoTracker Green and tetramethylrhodamine methyl ester (TMRM) labeling was analyzed by flow cytometry. MitoTracker Green measures mitochondrial abundance while TMRM labels active mitochondria with intact membrane potential (*Floryk and Houstěk, 1999*). An alteration in mitochondrial abundance or active mitochondria was not observed in Mb1-cre ABCB7 cKO pro-B cells as MitoTracker Green and TMRM labeling were comparable to WT (*Figure 3D*). Additionally, Mb1-cre-ABCB7 cKO Fr. C cells did not have a difference in expression of VDAC1 (*Figure 3E*), an abundant anion channel in the outer mitochondrial membrane (*Colombini, 2004*). Because iron accumulation was occurring upon conditional deletion of ABCB7 in pro-B cells, elevated ROS may cause the observed block in B cell development. The dyes CellROX Green and MitoSOX Red were utilized to probe total cellular and mitochondrial ROS levels, respectively. Neither total cellular nor mitochondrial ROS were statistically different (*Figure 3F and G*), suggesting that there was not elevated ROS occurring in ABCB7-deficient pro-B cells.

To rule out ferroptosis, an iron-mediated form of regulated cell death, ABCB7-deficient pro-B cells were examined for the presence of lipid peroxides, a hallmark of ferroptotic cells (*Li et al., 2020*). To do so, Bodipy 581/591C-11 (Bodipy C-11) oxidation was assessed by flow cytometry. Bodipy C-11 emits red fluorescence until it is oxidized by lipid peroxides, after which the emission shifts to green fluorescence and can be used to quantify cells with increased lipid peroxides. There were no changes in oxidized Bodipy C-11 levels in Mb1-cre ABCB7 cKO pro-B cells (*Figure 3H*), indicating that there was not an increase of lipid peroxides characteristically found in ferroptotic cells. Additionally, intracellular glutathione (GSH) levels were analyzed utilizing the dye ThiolTracker Violet. GSH is an antioxidant utilized by cells to scavenge free radicals and other ROS, particularly lipid peroxides (*Haenen and Bast, 2014*; *Li et al., 2020*). Interestingly, a small but significant decrease in GSH levels was observed in ABCB7-deficient pro-B cells (*Figure 3I*). Together, these data demonstrate ABCB7-deficient pro-B cells accumulate iron, but this accumulation does not disrupt mitochondrial abundance or membrane potential, induce ferroptosis, or cause increased ROS formation.

## ABCB7-deficient pro-B cells are not undergoing elevated apoptosis

As there was no evidence of ferroptosis in ABCB7-deficient pro-B cells (*Figure 3G*), apoptosis was examined. Expression of the pro-survival factors Bcl-xL and Mcl-1 is critical for B cell development (*Grillot et al., 1996*; *Vikström et al., 2016*) and is not decreased in ABCB7-deficient Fr. C cells compared to WT Fr. C cells (*Figure 4A and B*). Bcl2 protein expression could not be detected in WT or ABCB7-deficient Fr. C cells (*Figure 4—figure supplement 1*; WT Fr. A cells were used as a positive

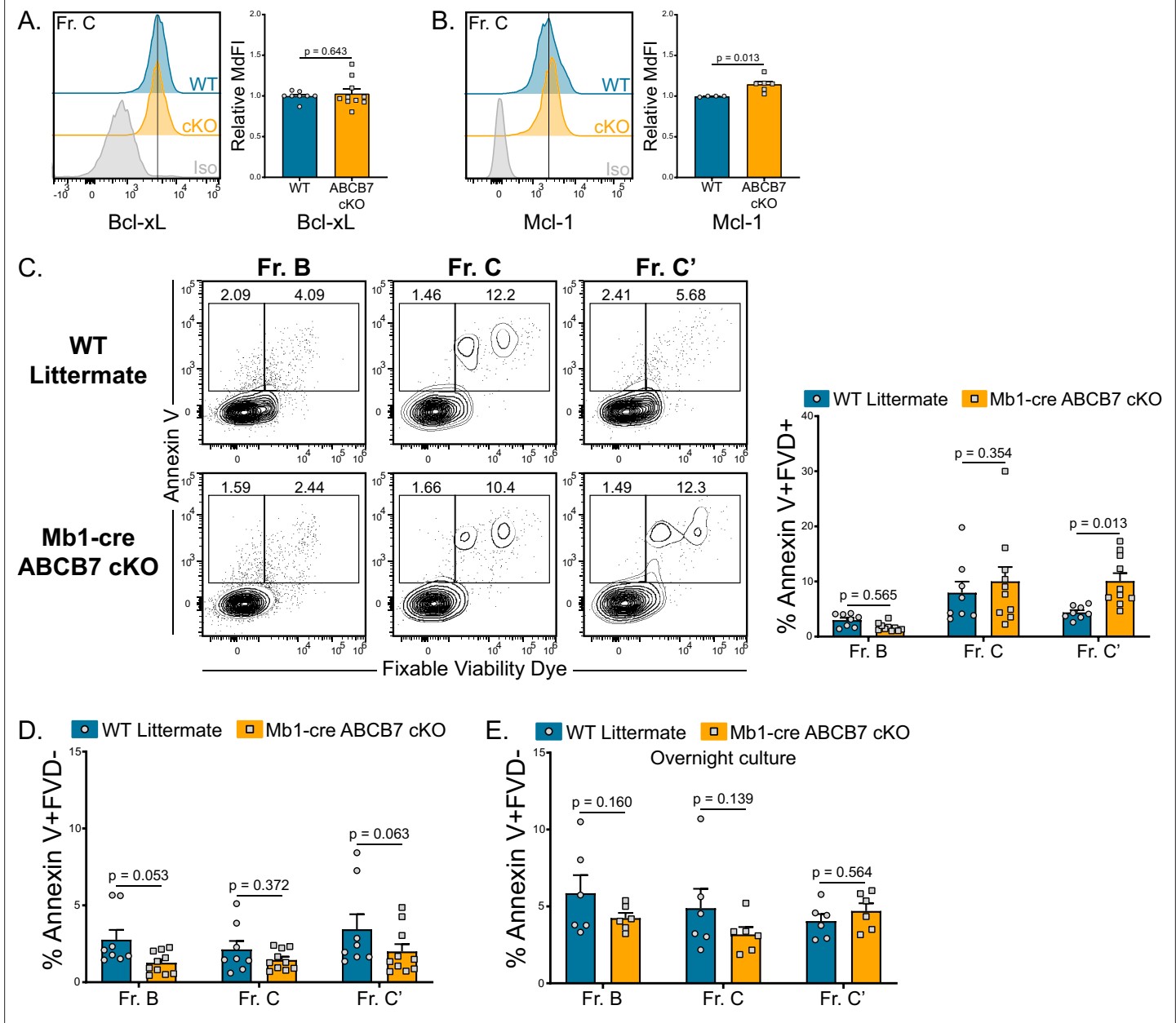

**Figure 4.** Analysis of apoptosis in ABCB7-deficient pro-B cells. (**A, B**) Intracellular flow cytometry analysis of Bcl-xL (**B**) and Mcl-1 (**C**) expression in Fr. C cells (B220+ CD19+ CD43+ BP-1+) from wild-type (WT) and Mb1-cre ABCB7 conditional knockout (cKO) mice. Quantification of MdFI is shown on the right of each plot. Isotype controls are shown in gray. Offset histograms are representative of at least three independent experiments (total of 4–10 mice/group). (**C**) Flow cytometry analysis of Annexin V binding and fixable viability dye (FVD) labeling of Fr. B-C' cells from WT and Mb1-cre ABCB7 cKO mice. Quantification of the proportion of each fraction that are dead (Annexin V+ FVD+) is shown on the right. Contour plots are representative of four independent experiments (total of 8–10 mice/group). (**D**) Quantification of the proportion of each fraction that are apoptotic (Annexin V+ FVD-) from (**C**). (**E**) Quantification of the proportion of each fraction that are apoptotic (Annexin V+ FVD-) after 16 hr in culture. Data represent three independent experiments (total of six mice/group). (**A–E**) Error bars represent SEM, and p-values are indicated above the data. Statistics were obtained by using an unpaired Student's t-test.

The online version of this article includes the following figure supplement(s) for figure 4:

**Figure supplement 1.** Bcl2 expression was undetected in pro-B cells.

control), a finding in line with previous literature (***Patton et al., 2014***; Immgen: ***Heng et al., 2008***). There was a small but significant increase in dead (Annexin V+ FVD+) ABCB7-deficient Fr. C' cells, but there was not a significant increase in dead pro-B cells (Fr. B-C) from Mb1-cre ABCB7 cKO mice

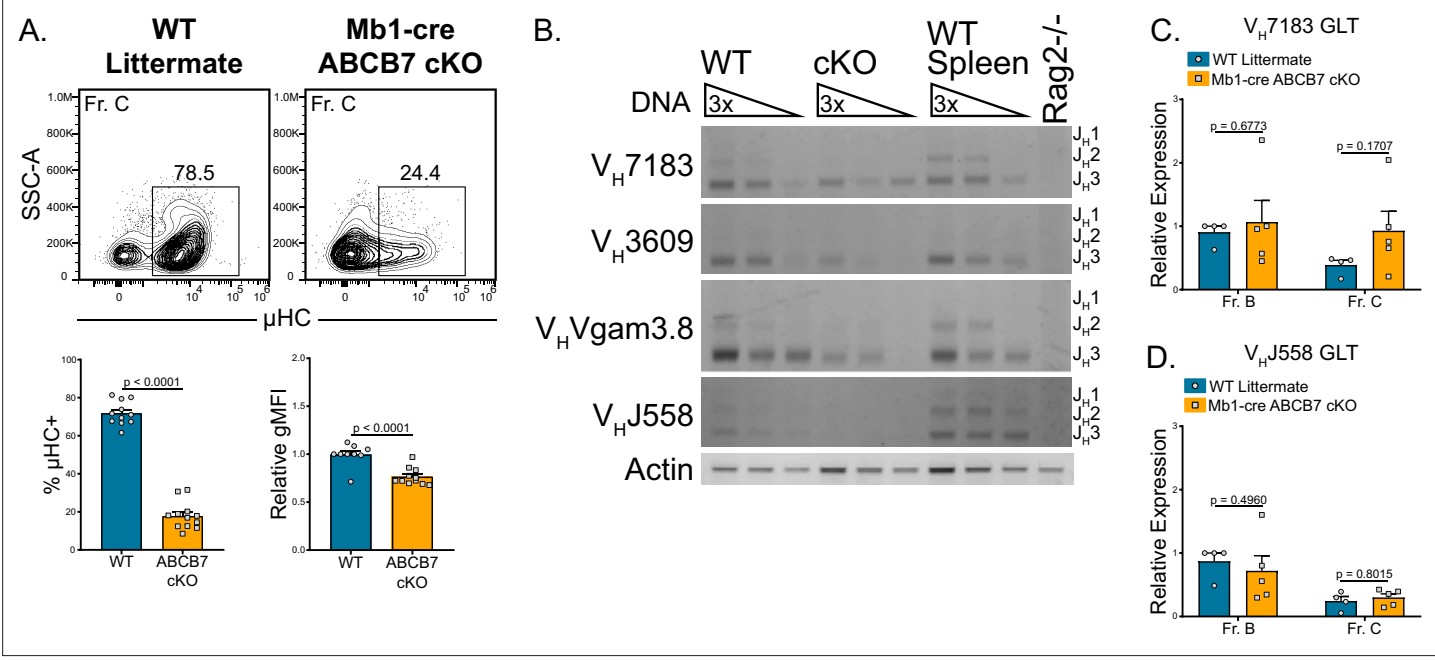

**Figure 5.** Reduced heavy chain recombination in ABCB7-deficient pro-B cells. (**A**) Intracellular flow cytometry analysis of μ heavy chain (μHC) expression in Fr. C cells (B220+ CD19+ CD43+ BP-1+) from wild-type (WT) and Mb1-cre ABCB7 conditional knockout (cKO) mice. Quantification of the proportion of μHC+ cells is shown on the left graph. Quantification of μHC gMFI is shown on the right graph. Contour plots are representative of six independent experiments (total of 12–15 mice/group). (**B**) Semiquantitative PCR analysis of heavy chain locus recombination. DNA was purified from magnetically enriched pro-B cells. DNA from magnetically enriched CD19+ WT splenocytes was used as a positive control, while DNA from a Rag2-deficient cell line was used as a negative control. DNA was adjusted to an equivalent concentration and subjected to threefold serial dilutions. Recombined $V_H$ gene segments were amplified using the indicated family-specific forward primer and a reverse primer specific to $J_H3$. Three bands corresponding to the usage of $J_H1$, $J_H2$, and $J_H3$ were expected for each $V_H$ gene amplified ($J_H1$ band is underrepresented due to product length). Results are ordered from proximal ($V_H7183$) to distal ($V_HJ558$) $V_H$ gene families. Actin was used as a loading control. Results are representative of four independent experiments. Image contrast and brightness were adjusted and colors were inverted for the final image. Source images are provided in *Figure 5—source data 1*. (**C**, **D**) Quantitative real-time PCR analysis of sterile $V_H7183$ (**C**) and $V_HJ558$ (**D**) germline transcript (GLT) expression in FACS sorted Fr. B (B220+ CD19+ CD43+ BP-1-) and Fr. C cells. *Hprt1* was used as an endogenous control, and relative expression values were normalized to expression in WT Fr. B cells. Results were obtained from three independent experiments (total of 4–5 mice/group). (**A**, **C**, **D**) Error bars represent SEM, and p-values are indicated above the data. Statistics were obtained using an unpaired Student's *t*-test.

The online version of this article includes the following figure supplement(s) for figure 5:

**Source data 1.** Source images.

---

(*Figure 4C*). There was also no difference in the proportion of apoptotic (Annexin V+ FVD-) pro-B cells from Mb1-cre ABCB7 cKO mice (*Figure 4D*). Apoptotic and dead pro-B cells are rapidly eliminated in the bone marrow (*Osmond et al., 1994*). To overcome this and determine if ABCB7-deficient pro-B cells were less viable, WT and Mb1-cre ABCB7 cKO bone marrow were placed in overnight cultures (16 hr) and Annexin V binding was assessed the next day. There was no difference in apoptosis of cultured pro-B cells from Mb1-cre ABCB7 cKO compared to WT (*Figure 4E*). This indicates that elevated apoptosis was not responsible for the reduced pro-B cell numbers observed in Mb1-cre ABCB7 cKO (*Figure 1C*), and together these data suggest that elevated apoptosis was not responsible for the block in B cell development in the bone marrow of Mb1-cre ABCB7 cKO mice.

## Reduced heavy chain recombination in ABCB7-deficient pro-B cells

The block in Mb1-cre ABCB7 cKO pro-B cell development (*Figure 1A and C*) suggested that altered recombination or expression of intracellular μHC was occurring in ABCB7-deficient pro-B cells. Indeed, a large reduction in the proportion of ABCB7-deficient Fr. C cells that expressed intracellular μHC as observed by flow cytometry (*Figure 5A*). To determine if this was due to a defect in recombination, a semiquantitative PCR assay was used to analyze recombination of the heavy chain locus in enriched Mb1-cre ABCB7 cKO pro-B cells. This PCR assay utilized 5′ primers specific for the $V_H7183$, $V_H3609$,

$V_H$Vgam3.8, or $V_H$J558 $V_H$ gene families and a 3' primer specific for the $J_H$3 gene (*Angelin-Duclos and Calame, 1998*; *Li et al., 1993*; *Pelanda et al., 2002*; *Schlissel et al., 1991*). Three product lengths were expected, depending on whether the VDJ recombination utilized $J_H$1, $J_H$2, or $J_H$3 genes ($J_H$1 products are underrepresented due to product size). WT pro-B cells had observable usage of each tested $V_H$ gene family (*Figure 5B*, left lanes). However, Mb1-cre ABCB7 cKO pro-B cells had a reduction in recombination of each of the $V_H$ gene families tested (*Figure 5B*, middle lanes). Additionally, the $V_H$ gene family usage that was observed in the ABCB7-deficient pro-B cells was skewed towards the more proximal $V_H$7183 gene family, while the most distal $V_H$J558 gene family did not have detectable usage (*Figure 5B*, middle lanes). These results indicated that heavy chain recombination was largely reduced and skewed towards proximal $V_H$ gene families upon conditional deletion of ABCB7 in pro-B cells. Expression of sterile germline transcripts (GLT) of $V_H$ genes has been used as a measure of locus accessibility in developing B cells (*Chen et al., 1993*; *Hesslein et al., 2003*), although production of GLT is not required for recombination (*Angelin-Duclos and Calame, 1998*). To determine if there was a difference in $V_H$ GLT and therefore locus accessibility, qPCR analysis of $V_H$7183 and $V_H$J558 GLT expression in sorted Fr. B and Fr. C cells from WT and Mb1-cre ABCB7 cKO mice was performed. Interestingly, expression of GLT for both $V_H$ gene families was found to be normal in both Fr. B and Fr. C cells (*Figure 5C and D*), suggesting that the reduction in heavy chain recombination was independent of locus accessibility. Collectively, these data suggest that heavy chain recombination and expression of μHC is reduced upon conditional loss of ABCB7 in developing pro-B cells and cannot be attributed to reduced locus accessibility.

## The MD4 HEL-Ig BCR transgene normalizes bone marrow B cell populations and restores splenic B cells in Mb1-cre ABCB7 cKO mice

Because decreased recombination or failure to express a rearranged heavy chain (*Figure 5A and B*) would cause the observed block in pro-B cell development (*Figure 1A and C*), Mb1-cre ABCB7 cKO mice were crossbred with mice bearing a transgenic, fully rearranged BCR specific to hen egg lysozyme (HEL; HEL-Ig; *Goodnow et al., 1988*; *Mason et al., 1992*). HEL-Ig WT and HEL-Ig Mb1-cre ABCB7 cKO mice had comparable proportions of CD19+ cells in the bone marrow (*Figure 6A*). However, absolute numbers of CD19+ cells (*Figure 6B*, left graph) were still reduced in the bone marrow of HEL-Ig ABCB7-deficient mice, despite normal proportions of CD19+ cells in these mice (*Figure 6B*, right graph). Importantly, splenic B cell proportions and cell numbers were equivalent between HEL-Ig WT and HEL-Ig Mb1-cre ABCB7 cKO mice (*Figure 6C and D*), showing that introduction of a fully rearranged BCR is able to restore splenic B cells in Mb1-cre ABCB7 cKO mice. Interestingly, HEL-Ig Mb1-cre ABCB7 cKO B cells from bone marrow (left plot) and spleen (right plot) had elevated intracellular iron as indicated by Phen Green quenching (*Figure 6E*). This demonstrated that elevated intracellular iron in ABCB7-deficient cells is not overtly toxic to mature B cells. Together, these data demonstrate that peripheral B cell numbers in Mb1-cre ABCB7 cKO mice can be restored upon introduction of a fully rearranged BCR.

## Reduced proliferation and evidence of DNA damage in ABCB7-deficient pro-B cells

One explanation for the reduction in heavy chain protein in ABCB7-deficient pro-B cells may be inability to repair double-stranded DNA breaks during VDJ recombination. Importantly, the Fe-S-GSH intermediates transported by ABCB7 mature into cofactors used in numerous DNA replication and damage repair enzymes including DNA primase, all replicative DNA polymerases, Dna2, FancJ, XPD, Endo III, and MutY (*Baranovskiy et al., 2018*; *Cunningham et al., 1989*; *Fuss et al., 2015*; *Klinge et al., 2007*; *Mariotti et al., 2020*; *Netz et al., 2011*; *Porello et al., 1998*; *Puig et al., 2017*; *Rudolf et al., 2006*). As recombination only occurs in nonproliferating cells due to Rag2 protein degradation (*Lin and Desiderio, 1994*), DNA damage was assessed by analyzing pH2A.X (γH2A.X) expression (*Smith et al., 2010*) in parallel with a 3 hr EdU pulse to identify proliferating pro-B cells. Strikingly, pH2A.X was highly expressed in EdU+ ABCB7-deficient Fr. B (B220+ CD19+ CD43+ BP-1-) and Fr. C (B220+ CD19+ CD43+ BP-1+) cells compared to EdU+ WT cells (*Figure 7A and B*). EdU- ABCB7-deficient cells did not have elevated expression of pH2A.X (*Figure 7A and B*), indicating that DNA damage was occurring in proliferating cells and was not due to heavy chain recombination. Analysis of EdU incorporation revealed that ABCB7-deficient Fr. B and Fr. C cells incorporated reduced amounts

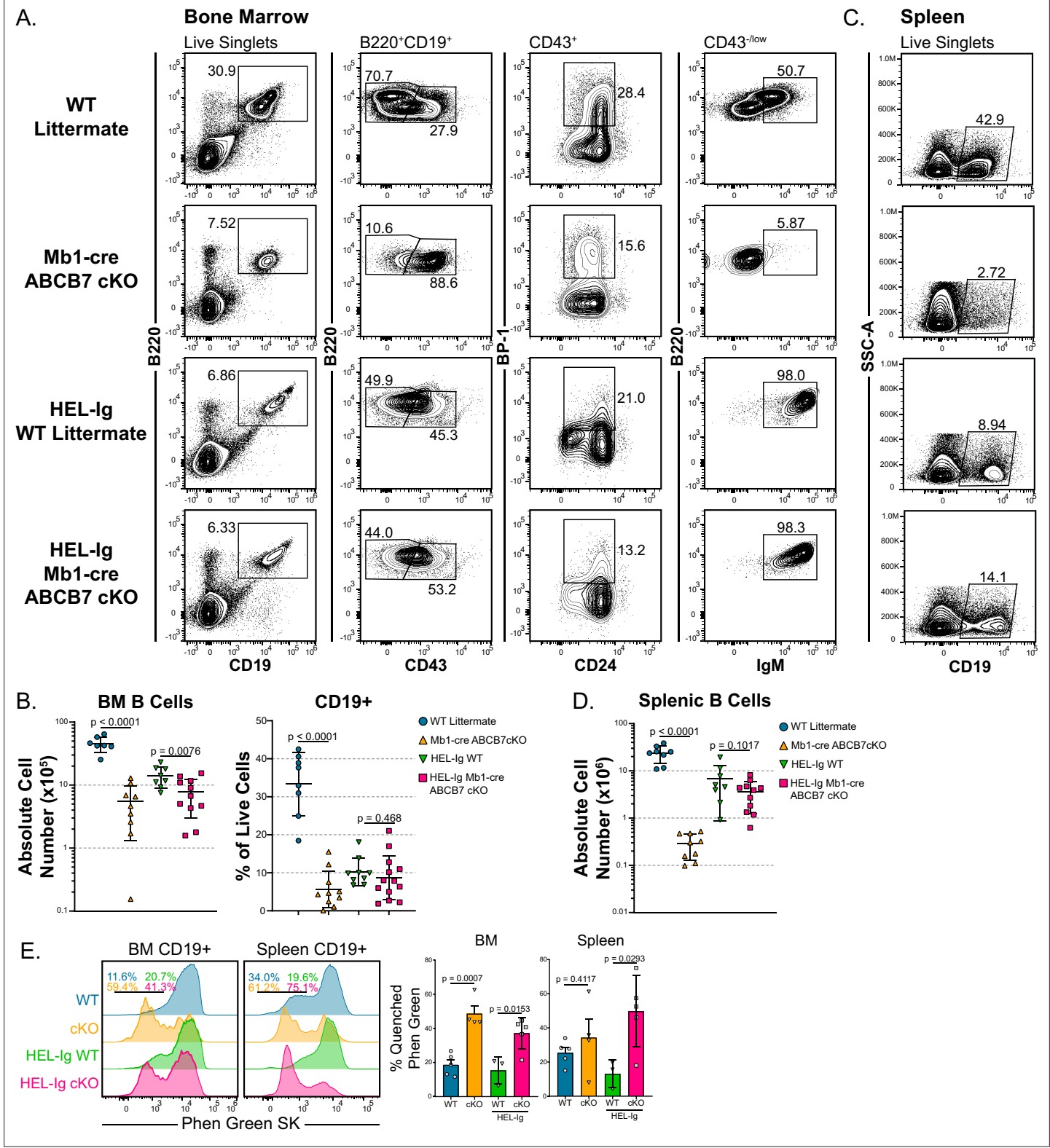

**Figure 6.** MD4 HEL-Ig transgenic B cell receptor (BCR) normalizes bone marrow B cell populations and restores splenic B cells in Mb1-cre ABCB7 conditional knockout (cKO) mice. (**A**) Flow cytometry analysis of B cell development in bone marrow from wild-type (WT) and Mb1-cre ABCB7 cKO mice in the presence or absence of a fully rearranged transgenic BCR specific for hen egg lysozyme (HEL-Ig). B cell populations were identified by gating on B220[+] CD19[+] cells: pro-B cells (CD43[+]), BP-1[+] pro-B cells, and Fr. E/F cells (CD43[-/low] IgM[+]). Contour plots are representative of seven independent experiments (total of 9–13 mice/group). (**B**) Graphs showing CD19[+] absolute cell numbers (left) and percentage of live cells (right) in the

*Figure 6 continued on next page*

Figure 6 continued

bone marrow of mice analyzed in (**A**). (**C**) Flow cytometry analysis of splenic CD19+ cells in mice from (**A**). Contour plots are representative of seven independent experiments (total of 9–13 mice/group). (**D**) Graph showing absolute cell numbers of CD19+ cells in the spleen of mice analyzed in (**C**). (**E**) Flow cytometry analysis of Phen Green SK fluorescence quenching by heavy metal ions in bone marrow and splenic CD19+ cells from WT and Mb1-cre ABCB7 cKO mice. Indicated values are the proportion of cells with quenched fluorescence, and quantification is shown on the right. Offset histograms are representative of three independent experiments (total of 3–5 mice/group). (**B, D, E**) Error bars represent SEM, and p-values are indicated above the data. Statistics were obtained by using an unpaired Student's *t*-test.

of EdU over the 3 hr pulse (*Figure 7C and D*), suggesting that these cells were undergoing slower DNA replication and reduced proliferation, consistent with induction of the S-phase DNA damage checkpoint (*Ciardo et al., 2019*). Because pH2A.X expression was elevated in ABCB7-deficient pro-B cells, expression of poly (ADP-ribose) polymerase (PARP), an important sensor of DNA damage that recruits repair enzymes (*Wang et al., 2019*), was analyzed and found to be significantly elevated in ABCB7-deficient Fr. C cells (*Figure 7E*). These data suggest that DNA damage-sensing pathways were active in ABCB7-deficient pro-B cells.

To prevent genomic instability during replication, the S-phase checkpoint slows replication in the presence of DNA damage (*Ciardo et al., 2019*). Checkpoint kinase 1 (Chk1) is an effector kinase that enforces the S-phase checkpoint until DNA damage is resolved and is required for B cell development at the pro- to pre-B cell transition (*Boddy et al., 1998*; *Feijoo et al., 2001*; *Schuler et al., 2017*). Analysis by flow cytometry revealed a slight but significant decrease in the expression of phosphorylated Chk1 (pChk1) in ABCB7-deficient Fr. C cells (*Figure 7F*). There was not a significant decrease in the expression of total Chk1 in these cells (*Figure 7G*). Intriguingly, this suggests that the activation of effector kinases during the S-phase checkpoint in ABCB7 deficient pro-B cells is partially diminished. The tumor suppressor p53 is a downstream target of the Chk1 during the S-phase checkpoint and is stabilized to control the cell cycle in the presence of DNA damage (*Shieh et al., 2000*). p53 expression was analyzed in ABCB7-deficient Fr. C cells and was also found to be significantly decreased in these cells (*Figure 7H*). To determine if DNA damage-sensing pathways were altered upstream of Chk1 activation, expression of phosphorylated ataxia-telangiectasia mutated (pATM), a checkpoint kinase upstream of Chk1 that is activated in the presence of DNA damage (*Smith et al., 2010*), was analyzed. Expression of pATM was significantly reduced in ABCB7-deficient Fr. C cells (*Figure 7I*), suggesting that the DNA damage response was less active in ABCB7-deficient pro-B cells, despite increased pH2A.X expression in proliferating cells.

Cyclin-dependent kinase 2 (CDK2) acts both upstream and downstream of Chk1 in the presence of DNA damage and strengthens the S-phase checkpoint. Loss of CDK2 expression delayed S/G2 progression in the presence of DNA damage and knockdown of CDK2 promoted cell cycle exit, including decreased expression of the proliferation-associated marker Ki-67 as well as Chk1 phosphorylation (*Bačević et al., 2017*). CDK2 expression was significantly decreased in ABCB7-deficient Fr. C cells (*Figure 7J*). Ki-67 was also analyzed in Fr. C cells from Mb1-cre ABCB7 cKO mice and was strikingly decreased compared to WT Fr. C cells (*Figure 7K*), suggesting that ABCB7-deficient cells had lost proliferation potential and were dropping out of the cell cycle in response to DNA damage occurring during DNA replication. Together, the reduction in CDK2 and loss of Ki-67 are consistent with an extended S-phase checkpoint. Cell cycle analysis using DAPI revealed that Fr. B cells from Mb1-cre ABCB7 cKO mice had a larger percentage of cells in G1 and a reduction in the percentage of cells in G2, while the proportion of cells in S phase was unchanged (*Figure 7L*, left-hand plot and graph). Fr. C cells had evidence that fewer cells were progressing through the cell cycle as more cells were in G1, and fewer cells were in S and G2 phases (*Figure 7L*, right-hand plots and graph). These data support the loss of proliferation potential in ABCB7-deficient pro-B cells as more cells are in G1 and fewer Fr. C cells are progressing through S and G2/M phases. Thus, ABCB7-deficient pro-B cells show evidence of replication-induced DNA damage, slower replication, and loss of proliferative capacity.

## ABCB7 is required for peripheral B cell proliferation and class switching

Cell proliferation has a well-characterized role in efficient CSR, and inhibition of proliferation can result in reduced class switching in activated B cells (*Hasbold et al., 1998*; *Hodgkin et al., 1996*; *Limon et al., 2014*; *Rush et al., 2005*; *Stavnezer et al., 2008*). Because evidence of reduced proliferation

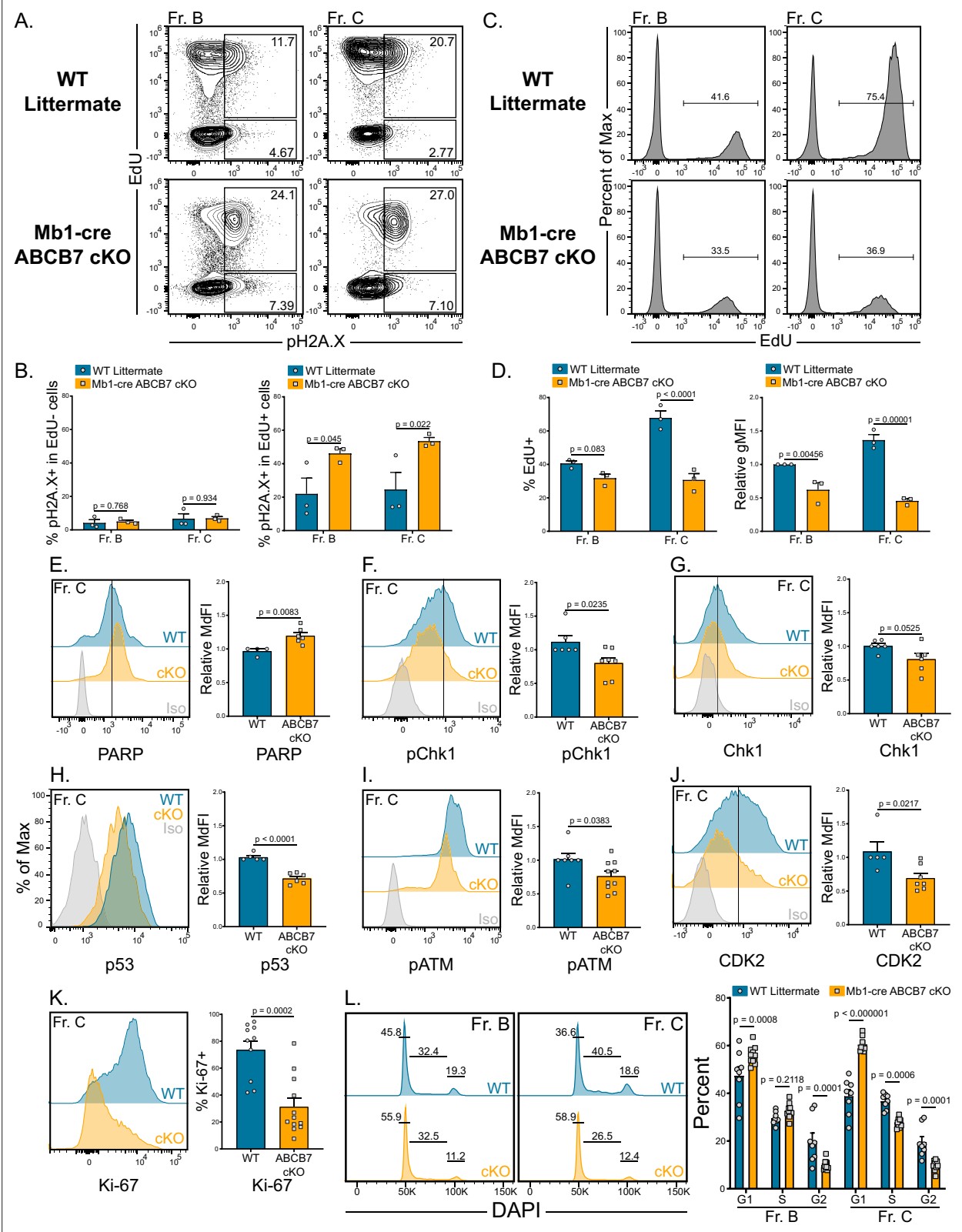

**Figure 7.** Reduced proliferation and evidence of DNA damage in ABCB7-deficient pro-B cells. (**A**) Intracellular flow cytometry analysis of EdU incorporation and pH2A.X expression in Fr. B (B220+ CD19+ CD43+ BP-1-) and Fr. C (B220+ CD19+ CD43+ BP-1+) cells from wild-type (WT) and Mb1-cre ABCB7 conditional knockout (cKO) mice. Cells were pulsed with EdU for 3 hr in culture. Contour plots are representative of three independent experiments (total of three mice/group). (**B**) Quantification of the proportion of EdU- cells (left graph) and EdU+ cells (right graph) that were positive

*Figure 7 continued on next page*

*Figure 7 continued*

for pH2A.X expression. (**C**) Flow cytometric analysis of the proportion of Fr. B and Fr. C cells (**A**) that incorporated EdU. Histograms are representative of three independent experiments (total of three mice/group). (**D**) Quantification of the proportion of cells that incorporated EdU (left plot) and EdU gMFI (right graph) in Fr. B and Fr. C cells from (**C**). gMFI was normalized to WT Fr. B cells. (**E–K**) Intracellular flow cytometry analysis of PARP (**E**), pChk1 (**F**), total Chk1 (**G**), p53 (**H**), pATM (**I**), and CDK2 (**J**), and Ki-67 (**K**) expression in Fr. C cells from WT and Mb1-cre ABCB7 cKO mice. Quantification of the MdFI or percent positive is shown on the right of each plot. Isotype controls are shown in gray. Offset and overlaid histograms are representative of at least three independent experiments (total of 4–10 mice/group). (**L**) Analysis of cell cycle status using intracellular DAPI staining in Fr. B and Fr. C cells from WT and Mb1-cre ABCB7 cKO mice. Leftmost gate marks cells in G1, middle gate marks cells in S phase, and rightmost gate marks cells in G2/M phases. Values shown above gates were derived from the FlowJo cell cycle analysis modeling tool. Quantification of the proportion of cells in G1, S, and G2/M phases is shown on the right of the plot. Proportions were determined by using the FlowJo cell cycle analysis modeling tool. Offset histograms are representative of six independent experiments (total of 8–10 mice/group). (**B, D–L**) Error bars represent SEM, and p-values are indicated above the data. Statistics were obtained by using an unpaired Student's *t*-test.

was observed in ABCB7-deficient pro-B cells, proliferation and CSR were examined in splenic B cells from CD23-cre ABCB7 cKO mice. To do so, enriched B220⁺ CD19⁺ B cells were cultured for 4 days with lipopolysaccharide (LPS) and various cytokines and/or anti-IgD dextran to induce proliferation and class switching to IgG1, IgG2a, IgG2b, IgG3, or IgA (see Materials and methods, *Guikema et al., 2010*). Despite normal proportions and numbers of splenic B cells in CD23-cre ABCB7 cKO mice, cells from these mice had a significant defect in class switching upon stimulation in culture (*Figure 8A and B*). While IgG2a- and IgA-stimulating cultures did not have a significant difference in the proportion of class-switched cells (*Figure 8B*, top graph), all conditions had a striking decrease in the number of cells that class switched in these cultures (*Figure 8B*, bottom graph). Interestingly, IgG1- and IgG2b-stimulating conditions had a more profound defect in class switching, both in proportion and numbers of class-switched cells (*Figure 8A and B*). This suggests that the severity of the defect in class switching in the absence of ABCB7 was dependent upon stimulation signals. No differences in the proportion of class-switched B cells were observed in the spleens of naïve, unchallenged mice (*Figure 8—figure supplement 1*).

Analysis of cell proliferation, which was quantified by CFSE dilution, revealed that B220⁺ CD19⁺ cells in each class switch culture condition had significant defects in proliferation in the absence of ABCB7 (*Figure 8C and D*). ABCB7-deficient B cells stimulated in these cultures underwent fewer cell divisions (quantified as proliferation index, *Figure 8D*, left graph) and a larger number of cells were undivided (*Figure 8D*, right graph). As seen with expression of antibody isotypes above, the severity of the defect in proliferation in the absence of ABCB7 was context dependent, with IgG1- and IgG2b-stimulating conditions having a stronger effect on proliferation (*Figure 8C and D*). Because ABCB7-deficient cells had less robust proliferation, cell viability was observed over time in IgG1-stimulating culture conditions. WT cells had a clear increase in cell viability after 2 days of stimulation, consistent with robust proliferation, which was not evident in CD23-cre ABCB7 cKO IgG1-stimulating cultures (*Figure 8—figure supplement 2A*, solid lines). Interestingly, unstimulated CD23-cre ABCB7 cKO B cells did not have altered viability compared to unstimulated WT cells (*Figure 8—figure supplement 2A*, dashed lines), suggesting that this difference in cell viability is only apparent in stimulated cells due to altered proliferation. The dtableecrease in ABCB7-deficient cell viability was observed in each culture condition tested, with ABCB7-deficient cells in IgG1- and IgG2b-stimulating cultures having a more severe decrease (*Figure 8—figure supplement 2B*) and a corresponding profound decrease in the number of live cells recovered (*Figure 8—figure supplement 2C*). Thus, peripheral B cells from CD23-cre ABCB7 cKO mice have reduced proliferation and cell viability. As reduced Ki-67 expression was observed in ABCB7-deficient pro-B cells, Ki-67 expression was analyzed in splenic B cells after 4 days in class switch culture conditions. Stimulated splenic B cells from CD23-cre ABCB7 cKO mice had a larger number of undivided, Ki-67⁻ cells compared to WT mice (*Figure 8E*), consistent with a loss of proliferation potential in the absence of ABCB7. Intriguingly, there were no differences in Ki-67 expression in T1, T2, T3, FO, or MZ B cells from the spleen of CD23-cre ABCB7 cKO mice analyzed ex vivo (*Figure 8—figure supplement 3*). These data suggest that the reduced cell proliferation after in vitro stimulation affected CSR in ABCB7-deficient peripheral B cells, and the severity of the defect was dependent upon stimulation signals received. Together, these data demonstrate that ABCB7 is essential for splenic B cell proliferation and class switch after activation.

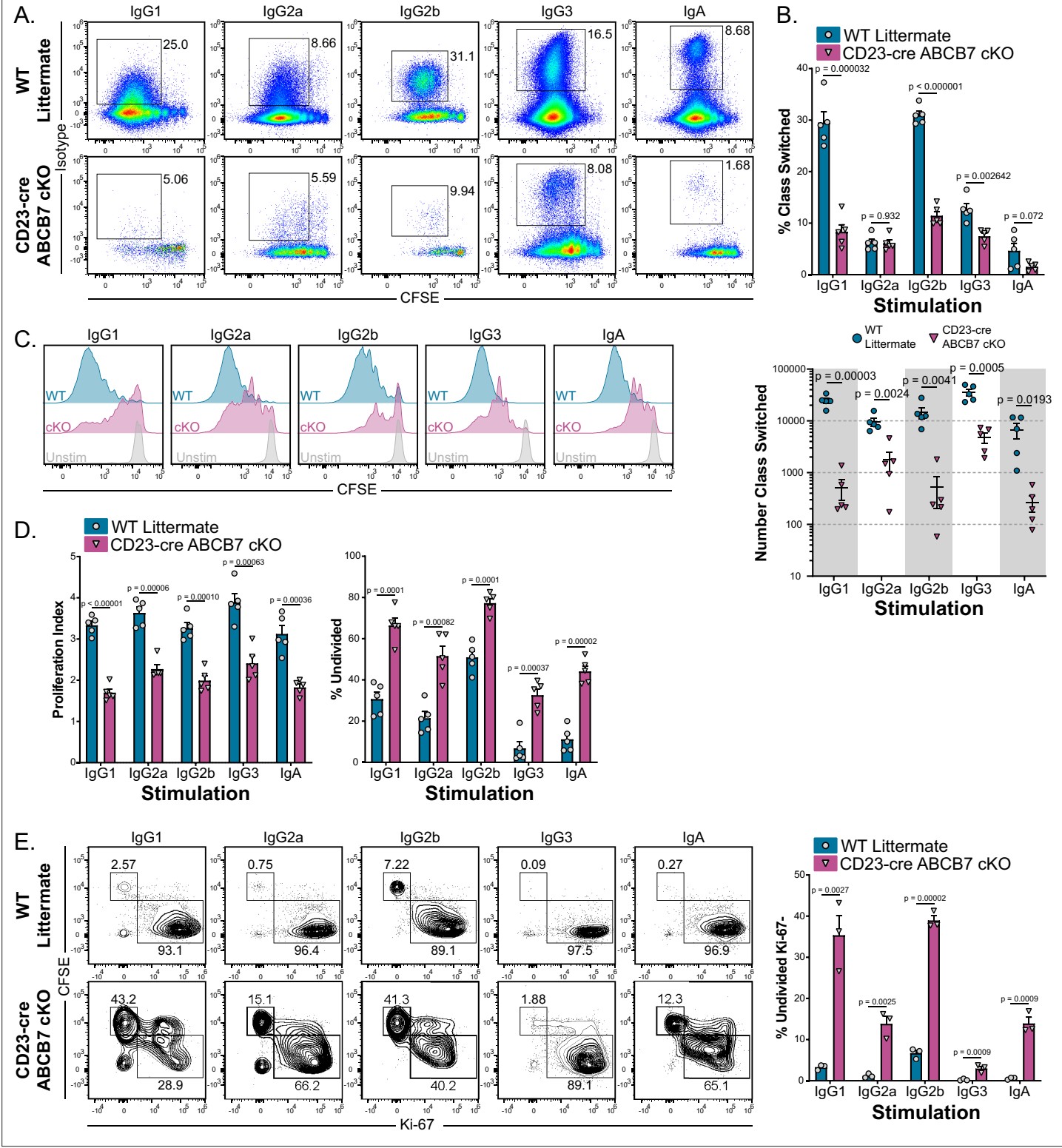

**Figure 8.** ABCB7 is required for peripheral B cell proliferation and class switching. (**A**) Flow cytometry analysis of IgG1, IgG2a, IgG2b, IgG3, and IgA expression on enriched B220[+] CD19[+] B cells from wild-type (WT) and CD23-cre ABCB7 conditional knockout (cKO) mice after 4 days in culture conditions that induce class switching to the indicated isotypes. Pseudocolor dot plots are representative of five independent experiments (total of five mice/group). (**B**) Quantification of the proportion (top) and number (bottom) of cells from (**A**) that class switched to the indicated antibody isotypes. The reported cell number was derived from flow cytometry live CD19[+] cells during analysis. (**C**) Flow cytometry analysis of carboxyfluorescein diacetate

*Figure 8 continued on next page*

*Figure 8 continued*

succinimidyl diester (CFSE) dilution in cells from (**A**). Offset histograms are representative of five independent experiments (total of five mice/group). (**D**) FlowJo proliferation modeling tool was used to quantify the proliferation index (left) and percentage of undivided cells (right) in cells from (**C**). (**E**) Intracellular flow cytometric analysis of Ki-67 expression in proliferating B220$^+$ CD19$^+$ cells after 4 days in culture conditions that induce class switching. Contour plots are representative of three independent experiments (total of three mice/group). Quantification of the percentage of undivided, Ki-67$^-$ cells is shown on the graph on the right. (**B, D, E**) Error bars represent SEM, and p-values are indicated above the data. Statistics were obtained by using an unpaired Student's *t*-test.

The online version of this article includes the following figure supplement(s) for figure 8:

**Figure supplement 1.** Class-switched B cells in the spleens of CD23-cre ABCB7 conditional knockout (cKO) mice.

**Figure supplement 2.** Cell viability in class switch assay.

**Figure supplement 3.** Ki-67 expression in ABCB7-deficient splenic B cells.

## Improved proliferation of B cells from HEL-Ig Mb1-cre ABCB7 cKO mice

ABCB7-deficient pro-B cells had reduced Ki-67 expression, fewer cells progressing through the cell cycle, and reduced EdU incorporation (*Figure 7*), suggesting that these cells have reduced proliferation potential and slower DNA replication. Additionally, these ABCB7-deficient pro-B cells had evidence of DNA damage in proliferating cells, but not in nonproliferative cells that would be undergoing heavy chain recombination (*Figure 7*). This indicated that nonproliferating, ABCB7-deficient pro-B cells undergoing heavy chain recombination were not accumulating DNA damage. ABCB7-deficient splenic B cells also had reduced proliferation, class switching, and Ki-67 expression upon stimulation in culture, but the severity of the defect was signal-dependent (*Figure 8*). Therefore, it was intriguing that peripheral B cell proportions and numbers were restored in the spleens of HEL-Ig Mb1-cre ABCB7 cKO mice (*Figure 6*). Interestingly, there was no difference in Ki-67 expression between HEL-Ig WT and HEL-Ig Mb1-cre ABCB7 cKO pro-B cells (*Figure 9A*). Additionally, analysis of cell cycle status using DAPI revealed that ABCB7-deficient HEL-Ig CD127$^+$ pro-B cells had equivalent proportions of cells in G1, S, and G2/M phases (*Figure 9B*). These data suggest that ABCB7-deficient pro-B cells in HEL-Ig Mb1-cre ABCB7 cKO mice have intact proliferation potential. Confirming this, the proportion of cells with EdU incorporation after a 3 hr pulse was equivalent between HEL-Ig WT and HEL-Ig Mb1-cre ABCB7 cKO pro-B cells (*Figure 9D*, left graph). Similar to ABCB7-deficient pro-B cells (*Figure 7*), ABCB7-deficient HEL-Ig pro-B cells had a significant reduction in EdU gMFI (*Figure 9D*, right graph), indicating that DNA replication in HEL-Ig Mb1-cre ABCB7 cKO pro-B cells was slowed in the absence of ABCB7. In addition, like Mb1-cre ABCB7 cKO pro-B cells, HEL-Ig Mb1-cre ABCB7 cKO pro-B cells had increased expression of pH2A.X in proliferating cells but not in nonproliferating cells (*Figure 9E and F*). These data demonstrate that in the presence of a fully rearranged BCR, ABCB7-deficient cells have restored proliferation potential and EdU incorporation. HEL-Ig mice bear a fully rearranged BCR, with μ and δ constant regions under endogenous control of the Eμ enhancer, that is expressed early during B cell development and pro-B cell development is altered in these mice (*Goodnow et al., 1988*). Therefore, the presence of BCR signals received in developing pro-B cells in HEL-Ig mice may rescue proliferation in the absence of ABCB7. Expression of IgM and IgD on WT, Mb1-cre ABCB7 cKO, HEL-Ig WT, and HEL-Ig Mb1-cre ABCB7 cKO B220$^+$CD19$^+$CD43$^+$ cells was analyzed by flow cytometry. As expected, IgM and IgD were not detected on WT and Mb1-cre ABCB7 cKO pro-B cells while their HEL-Ig counterparts had a clear increase in IgM expression and a slight increase in IgD expression (*Figure 9G*). Interestingly, HEL-Ig Mb1-cre ABCB7 cKO pro-B cells had a reduction in IgM expression and an increase in IgD expression compared to HEL-Ig WT pro-B cells (*Figure 9G*). Together, these data show that despite slowed proliferation and evidence of DNA damage in proliferating cells, HEL-Ig ABCB7-deficient pro-B cells are able to reconstitute the peripheral B cell compartment, which may be due to signals received at the pro-B cell stage through IgM and IgD.

## Discussion

Here, we demonstrate that ABCB7 is critical for bone marrow pro-B cell development and for proliferation and CSR in splenic B cells, but dispensable for peripheral B cell homeostasis. Mb1-cre ABCB7 cKO mice had notable iron accumulation and a severe block during pro-B cell development (*Figures 1*

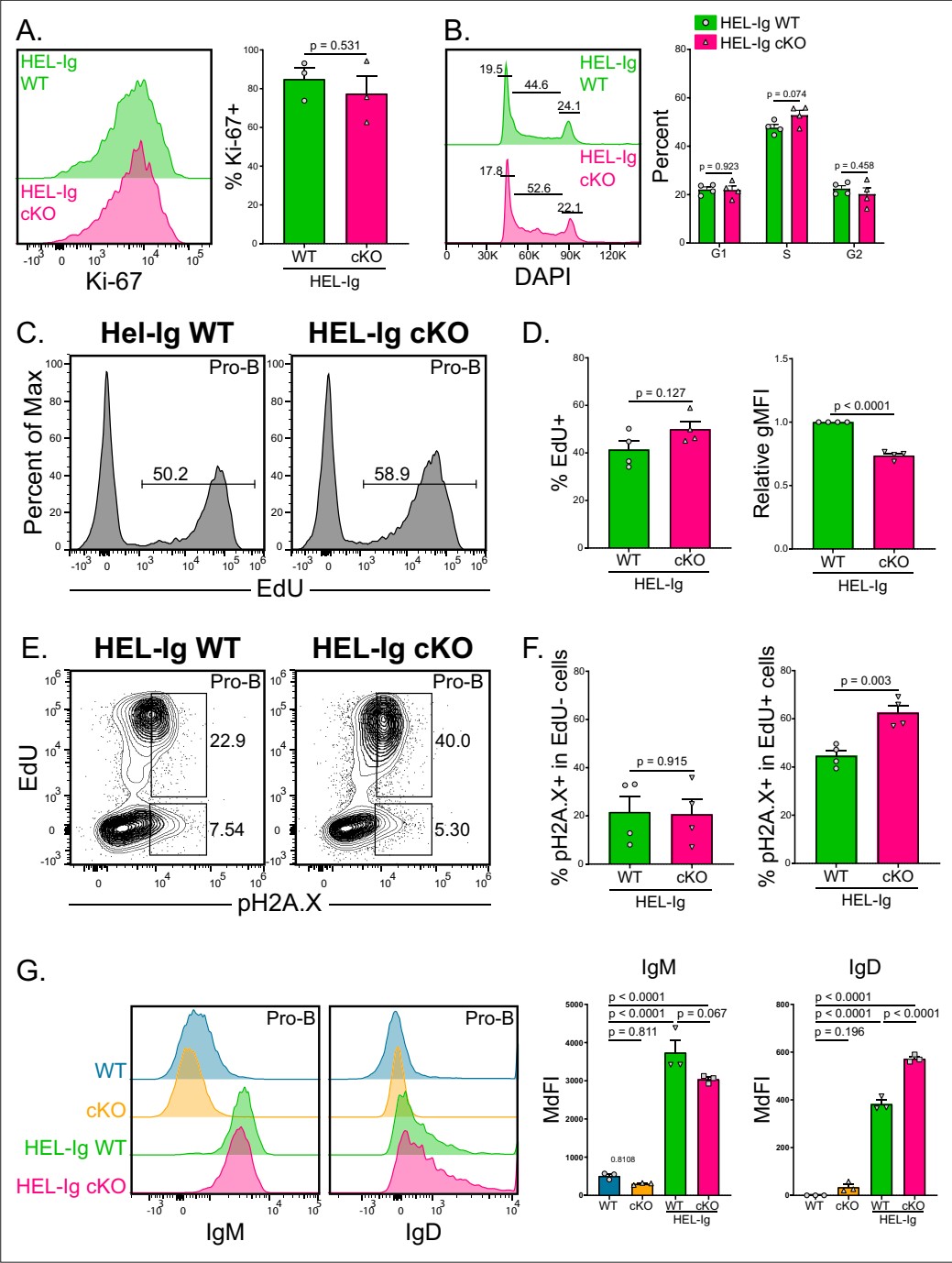

**Figure 9.** Improved proliferation of B cells from HEL-Ig Mb1-cre ABCB7 conditional knockout (cKO) mice. (**A**) Intracellular flow cytometry analysis of Ki-67 expression in pro-B cells (B220+ CD19+ CD43+ CD127+) from HEL-Ig wild-type (WT) and HEL-Ig Mb1-cre ABCB7 cKO mice. Quantification of the percent of Ki-67+ cells is shown on the right. Offset histogram is representative of three independent experiments (total of three mice/group). (**B**) Analysis of cell cycle status using intracellular DAPI staining in CD127+ pro-B cells from HEL-Ig WT and HEL-Ig Mb1-cre ABCB7 cKO mice. Leftmost gate marks cells in G1, middle gate marks cells in S phase, and rightmost gate marks cells in G2/M phases. Values shown above gates were derived from the FlowJo cell cycle analysis modeling tool. Quantification of the proportion of cells in G1, S, and G2/M phases is shown on the right of the plot. Proportions were determined by using the FlowJo cell cycle analysis modeling tool. Offset histograms are representative of four independent experiments (total of four mice/group). (**C**) Intracellular flow cytometric analysis of the proportion of pro-B cells from HEL-Ig WT and Hel-Ig Mb1-cre ABCB7 cKO mice that incorporated EdU. Cells were pulsed with EdU for 3 hr in culture. Histograms are representative of four independent experiments (total of four mice/

*Figure 9 continued on next page*

*Figure 9 continued*

group). (**D**) Quantification of the proportion of cells that incorporated EdU (left plot) and EdU gMFI (right plot) in pro-B cells from (**C**). (**E**) Intracellular flow cytometry analysis of pH2A.X expression in pro-B cells from (**C**). Contour plots are representative of four independent experiments (total of four mice/group). (**F**) Quantification of the proportion of EdU⁻ cells (left graph) and EdU⁺ cells (right graph) that were positive for pH2A.X expression. (**G**) Flow cytometry analysis of IgM (left) and IgD (right) expression in pro-B cells from WT, Mb1-cre ABCB7 cKO, HEL-Ig WT, and HEL-Ig Mb1-cre ABCB7 cKO mice. Quantification of the MdFI is shown on the right. Offset histograms are representative of three independent experiments (total of three mice/group). Error bars represent SEM, and p-values are indicated above the data. Statistics were obtained by using a one-way ANOVA with Tukey's multiple comparisons test. (**A, B, D, F**) Error bars represent SEM, and p-values are indicated above the data. Statistics were obtained by using an unpaired Student's *t*-test.

*and 3A*). Strikingly, ABCB7 was not required for splenic B cell homeostasis as CD23-cre ABCB7 cKO mice had normal populations and numbers of peripheral B cells (*Figure 1*). Surprisingly, splenic B cells deficient in ABCB7 did not exhibit iron accumulation (*Figure 3—figure supplement 1*). The block in pro-B cell development was not due to alterations in critical transcription factors, iron-related cellular stress, or elevated apoptosis (*Figures 2–4*). ABCB7-deficient cells had significantly reduced expression of intracellular µHC and diminished recombination at the heavy chain locus (*Figure 5*). These data suggested that ABCB7-deficient pro-B cells either had defective recombination or were halted in pro-B development prior to recombination.

Introduction of a fully rearranged transgenic BCR was able to restore bone marrow B cell proportions and splenic B cell numbers in Mb1-cre ABCB7 cKO, despite these cells still having iron accumulation (*Figure 6*). Analysis of proliferation using short-term EdU labeling demonstrated that fewer ABCB7-deficient pro-B cells incorporated EdU and that ABCB7-deficient pro-B cells incorporated less EdU compared to WT cells (*Figure 7C and D*). Interestingly, ABCB7-deficient pro-B cells that incorporated EdU had elevated expression of pH2A.X compared to WT pro-B cells (*Figure 7A and B*), which suggested an increase in DNA damage in proliferating cells but not cells undergoing heavy chain recombination. Pro-B cells from Mb1-cre ABCB7 cKO mice also had altered DNA damage sensing and a striking loss of proliferation potential as measured by Ki-67 expression (*Figure 7K*). Interestingly, pro-B cell proliferation was found to be restored in developing B cells bearing a fully rearranged transgenic HEL-Ig receptor, despite these cells still having evidence of elevated DNA damage in the absence of ABCB7 (*Figure 9*). This indicates that the defect in pro-B cell development in the absence of ABCB7 is likely due to a proliferation defect rather than simply an accumulation of DNA damage. How proliferation is restored in HEL-Ig Mb1-cre ABCB7 cKO pro-B cells remains to be seen. It may be due to the nature of the signal received by HEL-Ig pro-B cells as normally pro-B cells receive signals through the pre-BCR to pass the heavy chain checkpoint, but would receive signals through IgM and IgD in HEL-Ig transgenic mice instead (*Figure 9G*).

Although there was no defect in peripheral B cell homeostasis, ABCB7-deficient splenic B cells also had a significant defect in proliferation and Ki-67 expression during in vitro CSR assays (*Figure 8*), demonstrating that ABCB7 is critical for peripheral B cell proliferation as well. Intriguingly, some conditions in the CD23-cre ABCB7 cKO class switching cultures had a more profound defect in B cell proliferation and class switching, suggesting that different signaling pathways influence the severity of the proliferation defect. Together, these data demonstrate that ABCB7 is required for proliferation, pro-B cell development, and CSR.

Previous literature has demonstrated the importance of ABCB7 in iron homeostasis as some cell types accumulate mitochondrial iron, have defects in Fe-S cluster and heme synthesis, and have altered cytoplasmic aconitase activity upon the loss of ABCB7 (*Cavadini et al., 2007*; *Pondarré et al., 2006*). In agreement with these findings, we found that conditional deletion of ABCB7 in pro-B cells resulted in iron accumulation as indicated by Phen Green quenching (*Figure 3A*). Unexpectedly, we did not observe iron accumulation occurring in splenic B cells upon conditional deletion of ABCB7 in CD23-cre ABCB7 cKO mice at homeostasis. It has been hypothesized that other transporters may compensate for the loss of ABCB7, at least in some cell types. Liver-specific deletion of ABCB7 did not result in iron overload, which the authors suggested was due to unique iron homeostasis in hepatocytes or the existence of a complementary iron exporter (*Pondarré et al., 2006*). ABCB7 was also found to be dispensable for endothelial cells, although iron levels were never quantified in these cells (*Pondarré et al., 2006*). One candidate transporter that may share redundant function with ABCB7

is ABCB8, which is also thought to also transport Fe-S clusters to the cytoplasm. ABCB8-deficient cardiomyocytes displayed elevated levels of mitochondrial iron and ROS (*Ichikawa et al., 2012*). However, pro-B cells and follicular B cells express ABCB8 at similar levels (Immgen: *Heng et al., 2008*), suggesting that ABCB8 is likely not compensating for ABCB7-deficiency in CD23-cre ABCB7 cKO peripheral B cells at homeostasis or upon stimulation in class switch cultures. Cardiomyocytes also express ABCB7, which did not appear to compensate for loss of ABCB8 expression (*Ichikawa et al., 2012*; *Kumar et al., 2019*), implying that ABCB7 or ABCB8 have different roles in certain cell types. It is possible that a currently unknown iron exporter may compensate for the loss of ABCB7 in splenic B cells at homeostasis. Additionally, peripheral B cells may uniquely handle iron trafficking or storage compared to pro-B cells, which may prevent these cells from accumulating mitochondrial iron in the absence of ABCB7.

Iron overload is potentially toxic to cells as it can potently induce the formation of ROS, which can induce mitochondrial damage, disrupt the electron transport chain, or cause DNA damage (*Lawen and Lane, 2013*). Despite the extensive iron accumulation occurring in pro-B cells upon conditional deletion of ABCB7, we did not find evidence of elevated cellular or mitochondrial ROS (*Figure 3E and F*) and mitochondria were normal (*Figure 3D*). Two explanations for this are that the cells are effectively negating any excess ROS generated or the accumulating iron is being sequestered and stored in mitochondrial ferritin (*Lane et al., 2015*). We did observe reduced level of GSH (glutathione; *Figure 3H*), an abundant antioxidant utilized by cells to protect from ROS (*Haenen and Bast, 2014*), a possible indication that these cells are utilizing GSH to negate any excess ROS. However, ABCB7 transports an Fe-S-GSH intermediate (*Li and Cowan, 2015*), which may accumulate in the mitochondria and make GSH unavailable for antioxidant activity. It remains to be seen whether the lack of excess ROS and normal mitochondria is due to unique iron handling in pro-B cells and/or an effective antioxidant pathway protecting the cells from iron-derived ROS.

The Fe-S-GSH intermediates transported by ABCB7 mature in the cytoplasm where they are then used as critical cofactors for numerous enzymes involved in DNA replication and damage repair, including DNA primase, all replicative DNA polymerases, the helicases Dna2, FancJ, and XPD, and the glycosylases Endo III and MutY (*Baranovskiy et al., 2018*; *Cunningham et al., 1989*; *Fuss et al., 2015*; *Klinge et al., 2007*; *Mariotti et al., 2020*; *Netz et al., 2011*; *Porello et al., 1998*; *Puig et al., 2017*; *Rudolf et al., 2006*). Because we observed pH2A.X was highly expressed in EdU[+] cells, we hypothesize that DNA damage is being induced during replication due to defects caused by aberrant Fe-S cluster transport in the absence of ABCB7. The reduced amount of EdU[+] incorporation in Mb1-cre ABCB7 cKO pro-B cells implies that replication is slower in these cells (*Figure 7A–D*). HEL-Ig ABCB7-deficient pro-B cells also had increased pH2A.X expression in EdU[+] cells and had evidence of slower proliferation (*Figure 9C–F*), but these B cells were able to reconstitute peripheral B cell populations in the absence of ABCB7, implying that the slowed proliferation did not inhibit development of these cells. It is unclear if the slowed replication is due to slower polymerase, inefficient helicase-mediated DNA unwinding, increased DNA damage, or defective DNA damage repair, which are all possible if fewer Fe-S clusters are available for incorporation into critical enzymes. It is unlikely that any DNA damage occurring is due to excess ROS caused by iron accumulation because of the lack of excess ROS discussed above. DNA damage induced during replication may also explain the decrease in absolute numbers of Fr. B and Fr. C cells in Mb1-cre ABCB7 cKO mice as these cells undergo proliferation before undergoing heavy chain recombination (*Hardy et al., 1991*). Because DNA damage can block heavy chain recombination (*Arya and Bassing, 2017*; *Bahjat and Guikema, 2017*; *Fisher et al., 2017*), replication-induced DNA damage occurring in early B cell progenitors may also account for the reduction in heavy chain recombination and expression of μHC we observed in ABCB7-deficient pro-B cells (*Figure 5*).

The modest decrease in pATM, pChk1, and p53 (*Figure 7*) expression suggests that ABCB7-deficient cells also have a partially defective DNA damage response. The reduced expression of Chk1 and pChk1 (*Figure 7F and G*) is of particular interest because Chk1-deficient B cells have a block in B cell development at the pro-B cell stage. Chk1-haploinsufficient B cells had elevated DNA damage and underwent cell cycle arrest (*Schuler et al., 2017*), similar to what we observed in ABCB7-deficient pro-B cells. In addition to decreased Chk1 and pChk1 expression in ABCB7-deficient pro-B cells, we observed decreased CDK2 and Ki-67 expression. CDK2 has non-redundant functions during the S-phase checkpoint that promotes activation of DNA damage response and phosphorylation of Chk1.

Elimination of CDK expression delays S/G2 progression after DNA damage. Additionally, knockdown of CDK2 promotes a cell cycle exit program, as marked by reduction in Ki-67 and decreased phosphorylation of Chk1 (*Bačević et al., 2017*), which is in agreement with our data (*Figure 7F and K*). These data suggest that ABCB7-deficient pro-B cells encounter DNA damage during replication and either drop out of the cell cycle or are spending extended periods of time at the S-phase checkpoint. Stimulated ABCB7-deficient peripheral B cells also had a decrease in the proportion of cells expressing Ki-67 (*Figure 8E*), further suggesting a link between ABCB7 activity and proliferation potential as measured by Ki-67 expression. Interestingly, pro-B cells from HEL-Ig Mb1-cre ABCB7 cKO mice had restored Ki-67 expression, cell cycle status, and EdU incorporation, despite evidence of elevated DNA damage in proliferating ABCB7-deficient cells (*Figure 9A*). This suggests that different stimuli can influence proliferation in ABCB7-deficient cells, which is supported by the variable effect of ABCB7-deficiency on proliferation in different class switch conditions (*Figure 8*).

Currently, it is not clear why DNA damage is occurring in proliferating pro-B cells in the absence of ABCB7. It is also unclear why the DNA damage response is partially diminished in these cells. As mentioned, Fe-S clusters exported by ABCB7 are critical cofactors in numerous enzymes involved in DNA replication and damage repair. Whether Fe-S cluster incorporation into these enzymes is defective in the absence of ABCB7 remains to be seen. Additionally, it will be interesting to see if the activity of these DNA enzymes is diminished in ABCB7-deficient pro-B cells. Are pro-B cells more sensitive to alterations in the activities of these enzymes compared to peripheral B cells at steady state? And finally, is T cell development or homeostasis affected by the absence of ABCB7 or are pro-B cells uniquely sensitive to the loss of ABCB7? Future work exploring the role of ABCB7 in the development and homeostasis of lymphocytes will provide insight into how these cells regulate Fe-S export, iron homeostasis, proliferation, and DNA damage repair.

## Materials and methods

### Mice

The Institutional Animal Care and Use Committee at Mayo Clinic approved all animal studies performed in this article. $Abcb7^{fl}/Abcb7^{fl/fl}$(*Clarke et al., 2006*). Mb1-cre (*Hobeika et al., 2006*), MD4 HEL-Ig transgenic (*Goodnow et al., 1988*; *Mason et al., 1992*), and CD23-cre (*Kwon et al., 2008*) mice were all purchased from The Jackson Laboratory. Mb1-cre has *Cre* knocked into the *Cd79a* locus, replacing exons 2 and 3 (*Hobeika et al., 2006*). CD23-cre was generated by insertion of *Cre*, linked to a truncated human *Cd5* gene using an IRES, into exon 2 in a BAC clone containing the *Fcer2a* (*Cd23*) locus (*Kwon et al., 2008*). $Abcb7^{fl}$ and/or $Abcb7^{fl/fl}$ mice were interbred with Mb1-cre or CD23-cre mice to generate Mb1-cre ABCB7 cKO and CD23-cre ABCB7 cKO mice, respectively. HEL-Ig mice were crossbred with WT and Mb1-cre ABCB7 cKO mice to generate HEL-Ig WT and HEL-Ig Mb1-cre ABCB7 cKO mice, respectively. No differences were observed between male and female mice. All mice were housed in a barrier facility and were analyzed between the ages of 4–8 weeks for bone marrow experiments and between 8–12 weeks of age for splenic B cell experiments. For every experiment, age-matched littermate controls consisting of either ABCB7 floxed-only mice (with no Cre expression), Mb1-cre mice (with no floxed alleles), or wild-type (*Abcb7* WT) C57BL/6 mice were utilized, and for convenience these mice are referred to as simply 'WT littermate' in this article. Genotypes of all mice were confirmed by PCR after use.

### Cell lines

DNA from a RAG2$^{-/-}$ pro-B cell line was used as a negative control in the heavy chain semiquantitative PCR assay. The RAG2$^{-/-}$ pro-B cell line was previously established, maintained, and phenotyped by Dr. Medina (*Bertolino et al., 2005*; *Gwin et al., 2010*; *Pongubala et al., 2008*). These cells were validated as *Rag2*-deficient using qPCR. A mycoplasma test kit was used to confirm the absence of mycoplasma from the RAG2$^{-/-}$ pro-B cell line (ATCC, Manassas, VA).

### Preparation of single-cell suspensions

Single-cell suspensions of bone marrow were generated as previously described (*Amend et al., 2016*). Briefly, both femurs and tibias were dissected, cleaned of muscle tissue, and one end of each bone was snipped longitudinally about 2 mm using dissecting scissors to crack the bones open. Bones

were placed, cracked side down, into a 500 µL Eppendorf tube with a hole punched in the bottom using an 18G needle and then the smaller tube subsequently placed into a 1 mL Eppendorf tube. Bone marrow was spun out of the bones at 1500 rpm for 1 min and collected in the larger Eppendorf tube. Red blood cells were lysed with 1 mL of ACK Lysing Buffer (#118-156-101; Quality Biological, Gaithersburg, MD), subsequently diluted in 9 mL of PBS (#21-040-CMR; Corning, Corning, NY), and then filtered through an 80 µm Nylon mesh. Cells were centrifuged at 1500 rpm and washed twice with 10 mL of PBS. For preparation of single-cell suspension of splenocytes, spleens were dissected and homogenized between two frosted slides in 5 mL of PBS. After washing twice with 10 mL of PBS, red blood cells were lysed, and suspensions were diluted, filtered, and washed as above.

## Flow cytometry

All antibody dilutions, clones, and sources are provided in the Key resources table. Single-cell suspensions ($5 \times 10^6$ cells) from bone marrow or spleen were incubated (4°C, 10 min) with 5% mouse/rat serum (1:1) to block Fc receptors. For flow cytometric analysis of surface antigens, cells were incubated (4 °C, 30 min) with antibodies and fixable viability dye (FVD; Tonbo Biosciences, San Diego, CA). For analysis of bone marrow B cell populations, the following surface antibodies were used: anti-mouse B220 (clone RA3-6B2), anti-mouse BP-1 (Ly-51; clone BP-1 or 6C3), anti-mouse CD19 (clone 6D5), anti-mouse CD24 (clone 30-F1 or M1/69), anti-mouse CD43 (clone 1B11), and anti-mouse IgM (clone RMM-1). For analysis of splenic B cell populations, the following antibodies were used: anti-mouse AA4.1 (CD93; clone AA4.1), anti-mouse CD19 (clone 1D3), anti-mouse CD21/35 (clone 7E9), anti-mouse CD23 (clone B3B4), and anti-mouse IgM (clone RMM-1). The following surface antibodies were also used: CD2 (clone RM2-5), anti-human CD5 (CD23-cre reporter; clone L17F12), CD25 (clone PC61.5), anti-mouse CD71 (clone RI7217), anti-mouse CD127 (IL-7Rα; clone A7R34), and anti-mouse IgD (clone 11–26c.2a). The following antibodies were used for analysis of class switch isotypes: goat F(ab')$_2$ anti-mouse IgG1 (#1072-09), goat F(ab')$_2$ anti-mouse IgG2a (#1082-09), goat F(ab')$_2$ anti-mouse IgG2b (#1092-09), goat F(ab')$_2$ anti-mouse IgG3 (#1102-09), and goat anti-mouse IgA (#1040-09). For analysis of intracellular antigens, surface-stained cells were fixed and permeabilized using the FoxP3/Transcription Factor Staining Buffer Set (Tonbo Biosciences). Cells were incubated (4°C, 30 min) with 1× fixative, washed, and intracellular antibodies were incubated (4°C, 30 min) in 1× permeabilization buffer. The following intracellular antibodies were used: anti-Bcl-xL (#2767S), anti-mouse Bcl2 (#633508), anti-CDK2 (#14174), anti-Chk1 (ab32531), anti-mouse E47/E2A (#552510), anti-mouse EBF1 (ABE1294), anti-FOXO1 (#14262S), anti-mouse HO-1 (#ab69545), anti-mouse IgM (µHC; clone RMM-1; #406506), anti-mouse IKAROS (#89389S), anti-mouse IRF4 (#12-9858-82), anti-mouse Ki-67 (#652404 or #652411), anti-mouse Mcl-1 (#65617S), anti-p53 (#2015S), anti-PARP (#9532S), anti-mouse pATM (#651204), anti-mouse PAX5 (#17-9918-80), anti-mouse pChk1 (#13959S), anti-pH2A.X Ser139 (γH2A.X; #9720S), anti-mouse TdT (#12-5846-82), and anti-mouse VDAC1 (Porin; #55259-1-AP). Isotype control antibodies were included in experiments utilizing intracellular antibodies. All antibodies were purchased from Abcam (Cambridge, UK), BD Biosciences (Franklin Lakes, NJ), BioLegend (San Diego, CA), Cell Signaling Technology (Danvers, MA), eBioscience (Thermo Fisher; Waltham, MA), Millipore Sigma (Burlington, MA), ProteinTech (Rosemont, IL), SouthernBiotech (Birmingham, AL), or Tonbo Biosciences. Unless otherwise noted in figure legends, pro-B cells were defined as B220$^+$ CD19$^+$ CD43$^+$. Hardy fractions were defined as follows: Fr. B (B220$^+$ CD19$^+$ CD43$^+$ sIgM$^-$ BP-1$^-$), Fr. C (B220$^+$ CD19$^+$ CD43$^+$ sIgM$^-$ CD24$^{lo}$ BP-1$^+$ or B220$^+$CD19$^+$CD43$^+$sIgM$^-$BP-1$^+$, as denoted in figure legends), Fr. C' (B220$^+$ CD19$^+$ CD43$^+$ sIgM$^-$ CD24$^{hi}$ BP-1$^+$), Fr. D (B220$^+$ CD19$^+$ CD43$^{+/lo}$ sIgM$^-$), Fr. E (B220$^+$ CD19$^+$ CD43$^{+/lo}$ sIgM$^+$), and Fr. F (B220$^{hi}$ CD19$^+$ CD43$^{+/lo}$ sIgM$^+$). Peripheral B cells were defined as follows: T1 (CD19$^+$ AA4.1$^+$ CD21/35$^-$ IgM$^+$ CD23$^-$), T2 (CD19$^+$ AA4.1$^+$ CD21/35$^-$ IgM$^+$ CD23$^+$), T3 (CD19$^+$ AA4.1$^+$ CD21/35$^+$ IgM$^+$), FO (CD19$^+$ AA4.1$^-$ CD21/35$^+$ IgM$^+$), and MZ (CD19$^+$ AA4.1$^-$ CD21/35$^{hi}$ IgM$^{hi}$). Data were collected with an Attune NxT flow cytometer (Thermo Fisher), and all experiments were analyzed using FlowJo software (v10.5.3 or v10.8.0). Unless otherwise noted, all analyses utilized doublet exclusion (forward scatter [FSC] height/FSC area), size exclusion (side scatter [SSC] area/FSC area), and dead cell exclusion (FVD$^+$). Quantitative expression data are presented as median fluorescence intensity (MdFI), unless expression is not normally distributed in which case data are presented as the geometric mean of the fluorescence intensity (gMFI) (*Cossarizza et al., 2017*).

## FACS sorting

Cell sorting for qPCR was performed on a BD FACSMelody Cell Sorter (BD Biosciences). For pro-B cell populations, single-cell bone marrow suspensions were stained with anti-mouse B220 BV510 (1:200), anti-mouse BP-1 PE (1:50), anti-mouse CD19 PE-Cy7 (1:500), anti-mouse CD24 FITC (1:1000; clone 30-F1), anti-mouse CD43 PerCP (1:100), FVD Ghost Violet 450 (1:1000), and anti-mouse IgM PE-CF594 (1:100). Fr. B cells were gated as $FVD^-$ $B220^+$ $CD19^+$ $CD43^+$ $IgM^-$ $BP-1^-$. Fr. C cells were gated as $FVD^-$ $B220^+$ $CD19^+$ $CD43^+$ $IgM^-$ $CD24^+$ $BP-1^+$. For splenic B cell populations, single-cell splenocyte suspensions were stained with anti-mouse CD1d (1:100), anti-mouse CD19 eFluor 450 (1:500), anti-mouse CD21/35 PerCP-Cy5.5 (1:100), anti-mouse CD93 PE-Cy7 (1:100), FVD Ghost Violet 510 (1:1000), and anti-mouse IgM PE (1:100). FO B cells were gated as $FVD^-$ $CD19^+$ $AA4.1^-$ $CD21/35^+$ $IgM^+$. MZ B cells were gated as $FVD^-$ $CD19^+$ $AA4.1^-$ $CD21/35^{hi}$ $IgM^{hi}$ $CD1d^+$. Cells were sorted at 4°C into PBS and immediately used for RNA extraction.

## RNA purification, cDNA synthesis, and quantitative PCR

B cells were FACS sorted as described above. Sorted cells were lysed using QIAshredder spin-columns (Qiagen, Hilden, Germany) and total RNA was then extracted and purified using a RNeasy Mini Kit (Qiagen), both according to the manufacturer's instructions. Purified RNA was eluted from the columns using RNase-free water. cDNA was synthesized from purified RNA with random hexamers using a SuperScript IV First-Strand Synthesis System kit (Thermo Fisher) according to the manufacturer's instructions. After cDNA synthesis, RNA was removed using RNase H, as described in the SuperScript IV protocol. For analysis of *Rag1* and *Rag2* expression in Fr. B and Fr. C cells, cDNA was subjected to qPCR analysis using TaqMan probes specific for *Rag1* and *Rag2*. For analysis of *Abcb7* in FO and MZ B cells, cDNA was subjected to qPCR analysis using a TaqMan probe specific for *Abcb7*. Expression was normalized to that of an 18S rRNA TaqMan probe, and then normalized to expression in WT Fr. B cells or WT FO B cells. For analysis of $V_H7183$ and $V_HJ558$ GLT expression in pro-B cells, cDNA was subjected to qPCR analysis using SYBR Green and forward and reverse primers specific for a non-coding region of either $V_H7183$ or $V_HJ558$, as previously described (*Fuxa et al., 2004*). Primers used are listed in Appendix 2. Expression of each GLT was normalized to the expression of the housekeeping gene *Hprt* and then normalized to expression in WT Fr. B cells. The *Hprt* primers are listed in Appendix 2. For qPCR assays, every sample was plated in triplicate as a technical replicate. All qPCR assays were performed on a StepOne Real-Time PCR System (Thermo Fisher) and analyzed using the delta-delta Ct (ΔΔCt) method. All primers were ordered from Integrated DNA Technologies (Coralville, IA), and all qPCR probes were ordered from Thermo Fisher.

## Pre-B CFU assay

To enumerate IL-7-dependent pre-B CFU in bone marrow cells, MethoCult M3630 media (STEMCELL Technologies, Vancouver, Canada) was utilized according to the manufacturer's instructions. Briefly, single-cell suspensions of total bone marrow were prepared as described above and diluted in IMDM media containing 2% FBS to a concentration of $1 \times 10^6$ and $2 \times 10^6$ cells/mL. Two plating concentrations were used to account for seeding variability, as recommended by the manufacturer. To prepare the final culture concentrations, 400 μL of the cell suspensions were then added to 4 mL of the M3630 media to create a final cell concentration of $1 \times 10^5$ and $2 \times 10^5$ cells/mL. Samples were vigorously pulsed on a vortex. Bubbles were allowed to float for 5 min before 1.1 mL of the sample was drawn with a 16G needle syringe and distributed to the center of 35 mm dishes, in triplicate for each sample. Each sample dish was placed in a square 100 mm dish (with lid) along with one unlidded 35 mm dish containing water to maintain humidity. Cells were incubated for 8 days at 37°C in a 5% $CO_2$ incubator. After 8 days, pre-B cell colonies were counted as described in the manufacturer's protocol, and colony numbers were averaged across the triplicate plates. Colony numbers from the $2 \times 10^5$ cell concentration are reported in the article.

## Analysis of mitochondria, iron accumulation, ROS, GSH, and lipid peroxides

For flow cytometric analysis of mitochondria, single-cell suspensions ($5 \times 10^6$ cells) from bone marrow were incubated with 100 nM MitoTracker Green FM (#M7514) and 100 nM TMRM (#T668). Intracellular iron was quantified by incubating bone marrow cells with 5 μM Phen Green SK (#P14313)

diacetate. Intracellular and mitochondrial ROS were detected by incubating bone marrow cells with 5 μM CellROX (#C10444) and 5 μM MitoSOX (#M36008) dyes, respectively. To detect the presence of lipid peroxides, bone marrow cells were incubated with 2 μM Bodipy 581/591C-11 (#D3861), which is specifically oxidized by lipid peroxides. GSH levels were quantified by incubating bone marrow cells with 4 μM ThiolTracker Violet (#T10095), which detects GSH. All dyes were purchased from Thermo Fisher. For labeling with each dye, dyes were diluted in PBS and incubated with cells for 30 min in a 37°C 5% $CO_2$ incubator. After incubation, cells were washed with PBS. Fc receptors were blocked with mouse/rat serum, and surface antigens and FVD were stained as described above. Because of fluorescence spillover from these dyes, a limited surface marker panel was utilized in these experiments: anti-mouse B220, anti-mouse CD19, anti-mouse CD43, and anti-mouse IgM. Pro-B cells were defined as B220$^+$ CD19$^+$ CD43$^+$ sIgM$^-$. Data were collected with an Attune NxT flow cytometer (Thermo Fisher).

## Annexin V binding

Single-cell suspensions ($5 \times 10^6$ cells) were blocked, and surface antigens and FVD were labeled as described above. Cells were then washed with 1× Annexin V binding buffer (BD Biosciences) before being incubated (4°C, 15 min) with Annexin V-FITC conjugate (1:500; BD Biosciences) diluted in 1× binding buffer. After incubation, cells were washed with and resuspended in 1× binding buffer for immediate analysis on an Attune NxT flow cytometer (Thermo Fisher). Cell populations were gated without live/dead exclusion in order to visualize Annexin V$^+$ FVD$^+$ cells as presented in the article.

## Overnight pro-B cell culture

Single-cell suspensions of total bone marrow were prepared as described above. Cells were resuspended in culture media (IMDM, 10% FBS, 1% glutamine, 1% Pen/Strep, and 0.1% 2-mercaptoethanol [2-ME]) and $5 \times 10^6$ cells were placed in 6-well plates and incubated overnight for 16 hr at 37°C in a 5% $CO_2$ incubator. The next day, cells were washed with PBS and Annexin V binding was analyzed as described above.

## DNA purification from magnetically enriched pro-B cells

Single-cell suspensions of total bone marrow were prepared as described above. Pro-B cells were then enriched using an EasySep Mouse Streptavidin RapidSpheres Isolation Kit (STEMCELL Technologies) according to the manufacturer's instructions. The following biotinylated antibodies were used for negative selection of unwanted cells: CD11b (1:100), CD11c (1:100), CD4 (1:100), CD8 (1:500), GR-1 (1:100), IgM (1:100), NK1.1 (1:100), TCRγδ (1:100), and TCRβ (1:100). Note that IgM was included to eliminate IgM$^+$ pre-B, naïve, and recirculating B cells. For a positive control, splenic B cells were harvested from a WT mouse and subjected to a similar RapidSphere negative selection that did not include IgM antibodies. Purity was checked by using flow cytometry after magnetic separation. Genomic DNA was then isolated from enriched pro-B cells and splenic B cells using a DNeasy Blood and Tissue Kit (Qiagen) according to the manufacturer's instructions. The concentration of DNA was determined using a Nanodrop spectrophotometer (Thermo Fisher) and adjusted so that all samples would have equivalent concentrations.

## Semiquantitative PCR analysis of heavy chain recombination

Analysis of heavy chain recombination was performed as previously described (*Angelin-Duclos and Calame, 1998*; *Li et al., 1993*; *Pelanda et al., 2002*; *Schlissel et al., 1991*). Briefly, genomic DNA from magnetically enriched WT and Mb1-cre ABCB7 cKO pro-B cells (described above) was adjusted to equivalent concentrations and then underwent threefold serial dilutions. Serially diluted DNA was then subjected to a PCR assay that detects recombination between specified V$_H$ gene families and the J$_H$3 gene. This assay utilizes forward primers specific to either V$_H$7183, V$_H$3609, V$_H$Vgam3.8, or V$_H$J558 gene families and a reverse primer specific to the J$_H$3 gene (*Angelin-Duclos and Calame, 1998*; *Li et al., 1993*; *Pelanda et al., 2002*; *Schlissel et al., 1991*). Three PCR products were expected depending on whether the recombination utilized the J$_H$1, J$_H$2, or J$_H$3 genes; however, the largest product length is underrepresented due to product length. The primers used are listed in Appendix 2. Primers specific to actin were used as a loading control for each serial dilution and are listed in Appendix 2. DNA isolated from splenic B cells was used as a positive control for recombination events.

DNA from a Rag2$^{-/-}$ pro-B cell line was utilized as a negative control. PCR was performed using OneTaq DNA polymerase and buffers for GC-rich DNA (#M0480L; New England Biolabs, Ipswich, MA). Each PCR reaction additionally included 200 µM dNTPs, and forward and reverse primers at 0.5 µM (actin) or 1 µM ($V_H$ and $J_H3$ genes). For detection of actin, the reactions were incubated as follows: 95°C for 3 min, then 30 cycles of 95°C for 30 s, 55°C for 30 s, and 72°C for 60 s, and a final incubation at 72°C for 3 min. For detection of $V_H7183$ and $V_HJ558$ recombination, the reactions were incubated as follows: 95°C for 3 min, then 45 cycles of 95°C for 45 s, 63°C for 45 s, and 72°C for 60 s, and a final incubation at 72°C for 15 min. For detection of $V_H3609$ and $V_HVgam3.8$ recombination, the reactions were incubated as follows: 95°C for 3 min, then 45 cycles of 95°C for 45 s, 60°C for 60 s, and 72°C for 60 s, and a final incubation at 72°C for 10 min. PCR products were run on 1.25% agarose gels containing ethidium bromide and photographed using an Omega Lum G gel imager (Gel Company, Ramsey, MN). All PCR primers were ordered from Integrated DNA Technologies.

## EdU assay

EdU incorporation was analyzed using a Click-iT Plus EdU Alexa Fluor 488 Flow Cytometry Assay Kit (Thermo Fisher). Briefly, single-cell suspensions of bone marrow cells were prepared as described above. $1 \times 10^7$ cells were resuspended in complete media (RPMI, 10% FBS, 1% glutamine, 1% HEPES, 1% non-essential amino acids, 1% Pen/Strep, and 0.1% 2-ME) and plated in a 6-well plate. EdU was added to each well at a final concentration of 10 µM. A control well without EdU addition was also plated. Cells were incubated at 37°C for 3 hr in a 5% $CO_2$ incubator. After incubation, $3 \times 10^6$ cells were blocked (4°C, 10 min) with mouse/rat serum and surface antigens and FVD were labeled (4°C, 30 min) in PBS containing 1% BSA, as recommended by the manufacturer's instructions. Cells were washed and then incubated (room temperature, 15 min) with EdU kit fixative. Cells were then washed with PBS containing 1% BSA and then permeabilized by incubating (room temperature, 15 min) with 1× EdU kit permeabilization buffer. Click-iT reaction cocktails were then prepared according to the manufacturer's instruction. 500 µL of reaction cocktail was then added to each sample and incubated at room temperature for 30 min. After the Click-iT reaction, cells were then washed with 1× permeabilization buffer and incubated (4°C, 30 min) with intracellular antibodies specific for pH2A.X in 1× permeabilization buffer. Samples were washed with 1× permeabilization buffer and immediately analyzed on an Attune NxT flow cytometer (Thermo Fisher).

## DAPI staining

For analysis of cell cycle status, single-cell suspensions ($5 \times 10^6$ cells) from bone marrow were stained with surface antibodies and FVD and were then fixed and permeabilized using FoxP3/Transcription Factor Staining Buffer Set, as described above. Cells were stained with DAPI (1:4000) by incubating (4°C, 30 min) with DAPI in 1× permeabilization buffer. Samples were immediately analyzed on an Attune NxT flow cytometer (Thermo Fisher). The FlowJo cell cycle analysis tool was used to quantify the proportion of cells in each cell phase.

## Class switch culture

Single-cell suspensions of splenocytes were prepared as described above. Splenic B cells were then enriched using an EasySep Mouse Streptavidin RapidSpheres Isolation Kit (STEMCELL Technologies) according to the manufacturer's instructions. The following biotinylated antibodies were used for negative selection of unwanted cells: CD11b (1:100), CD11c (1:100), CD4 (1:100), CD8 (1:500), GR-1 (1:100), NK1.1 (1:100), TCRγδ (1:100), and TCRβ (1:100). After selection, B cells were labeled with 2.5 µM carboxyfluorescein diacetate succinimidyl diester (CFSE). Cells were cultured at $2.5 \times 10^5$/mL in a 24-well plate and stimulated to class switch to different antibody isotypes as described previously (*Guikema et al., 2010*). In detail, cells were cultured in complete media (RPMI, 10% stem cell-grade FBS, 1% glutamine, 1% HEPES, 1% non-essential amino acids, 1% Pen/Strep, and 0.1% 2-ME). Note that stem cell-grade FBS of the same lot (#10437028; Thermo Fisher) was utilized in each culture experiment as traditional FBS was found to inhibit class switching (a finding that was previously reported [*Zaheen and Martin, 2010*]). All culture conditions contained LPS (25 µg/mL; Millipore Sigma) and recombinant human BAFF (100 ng/mL; PeproTech, Cranbury, NJ). For their respective wells, the following were added to induce switching to different isotypes: for IgG1 switching, recombinant mouse IL-4 (20 ng/mL; PeproTech) was added; for IgG2a switching, IFNγ (25 ng/mL; PeproTech)

was added; for IgG2b switching, TGF-β (2 ng/mL; PeproTech) was added; for IgG3 switching, anti-δ-dextran (3 ng/mL; Fina Biosciences, Rockville, MD) was added; for IgA switching, recombinant mouse IL-4 (20 ng/mL), TGF-β (2 ng/mL), anti-δ-dextran (3 ng/mL), and IL-5 (2 ng/mL; PeproTech) were added. Cells were cultured for 4 days at 37°C in a 5% $CO_2$ incubator. After culture, surface antigens and FVD were labeled for flow cytometry as described above. CFSE dilution and class switching were also analyzed using flow cytometry. Intracellular flow cytometry was utilized to analyze Ki-67 expression. For the assay analyzing cell death over time, aliquots of cells from an IgG1-stimulating culture were harvested each day and analyzed for FVD binding. The FlowJo proliferating modeling tool was used to quantify proliferation index and percent undivided after culture.

## Statistical analysis

Statistical methods used are listed in each figure legend. Each data point of a bar graph represents a single mouse. Unpaired Student's *t*-tests were used to compare quantifications of MdFI, gMFI, proportions, expression, relative expression, and CFU colony counts between WT and Mb1-cre ABCB7 cKO or CD23-cre ABCB7 cKO mice, unless otherwise noted in figure legends. For comparison between WT, Mb1-cre ABCB7 cKO, and CD23-cre ABCB7 absolute cell numbers, a one-way ANOVA with Dunnett's test for multiple comparisons was utilized. A repeated measures two-way ANOVA with Geisser–Greenhouse correction and Holm–Šídák's multiple comparisons test were used for comparison between WT and CD23-cre ABCB7 cKO cell viability over time. A one-way ANOVA with Tukey's multiple comparisons test was used for comparison of IgM and IgD expression between WT, Mb1-cre ABCB7 cKO, HEL-Ig WT, and HEL-Ig Mb1-cre ABCB7 cKO pro-B cells. All error bars represent the mean ± SEM. p-Values are indicated on each figure and/or figure legend. Statistical analysis was performed using GraphPad Prism software.

## Acknowledgements

We thank the members of the VSS, KM, and Hu Zeng (Mayo Clinic) laboratories for their helpful discussions of this work.

## Additional information

### Funding

| Funder | Grant reference number | Author |
| --- | --- | --- |
| National Institute of Allergy and Infectious Diseases | 1R21 AI157328-01 | Virginia Smith Shapiro |
| National Institute of Allergy and Infectious Diseases | T32AI007425 | Michael Jonathan Lehrke |

The funders had no role in study design, data collection and interpretation, or the decision to submit the work for publication.

### Author contributions

Michael Jonathan Lehrke, Conceptualization, Data curation, Formal analysis, Funding acquisition, Investigation, Supervision, Writing - original draft, Writing – review and editing; Michael Jeremy Shapiro, Conceptualization, Data curation, Formal analysis, Investigation, Writing - original draft, Writing – review and editing; Matthew J Rajcula, Data curation, Formal analysis, Investigation, Resources, Writing – review and editing; Madeleine M Kennedy, Shaylene A McCue, Resources, Writing – review and editing; Kay L Medina, Conceptualization, Resources, Supervision, Writing – review and editing; Virginia Smith Shapiro, Conceptualization, Formal analysis, Funding acquisition, Supervision, Writing – review and editing

### Author ORCIDs

Michael Jonathan Lehrke http://orcid.org/0000-0002-2376-9168
Virginia Smith Shapiro http://orcid.org/0000-0001-9978-341X

### Ethics

This study was performed in strict accordance with the recommendations in the Guide for the Care and Use of Laboratory Animals of the National Institutes of Health. All of the animals were handled according to approved institutional animal care and use committee (IACUC) protocols (#A3738-18) of the Mayo Clinic.

### Decision letter and Author response

Decision letter https://doi.org/10.7554/eLife.69621.sa1
Author response https://doi.org/10.7554/eLife.69621.sa2

## Additional files

### Supplementary files

• Transparent reporting form

### Data availability

All data generated or analyzed during this study are included in the manuscript and supporting files. There are no large datasets included in this manuscript. Source data files have been provided for Figure 5.

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

# Appendix 1

**Appendix 1—key resources table**

| Reagent type (species) or resource | Designation | Source or reference | Identifiers | Additional information |
|---|---|---|---|---|
| Gene (*Mus musculus*) | Abcb7 | GenBank | GeneID:11306 | |
| Strain, strain background (*M. musculus*) | B6.129S4-*Abcb7*tm1Mdf/J (*ABCB7*fl/*ABCB7*fl/fl) | Ordered from The Jackson Laboratory, described in PMID:16424901 | Cat#:006490; MGI: 3628655; RRID:IMSR_JAX:006490 | ABCB7 floxed mice |
| Strain, strain background (*M. musculus*) | B6.C(Cg)-*Cd79a*tm1(cre)Reth/EhobJ (Mb1-cre) | Ordered from The Jackson Laboratory, described in PMID:16940357 | Cat#:020505; MGI: 3687451; RRID:IMSR_JAX:020505 | Bone marrow B cell-specific cre |
| Strain, strain background (*M. musculus*) | B6.Cg-Tg(*Fcer2a*-cre)5Mbu/J (CD23-cre) | Ordered from The Jackson Laboratory, described in PMID:18538592 | Cat#:028197; MGI: 3803652; RRID:IMSR_JAX:028197 | Peripheral B cell-specific cre, uses B cell-specific *Cd23* promoter |
| Strain, strain background (*M. musculus*) | C57BL/6-Tg(IghelMD4)4Ccg/J (HEL-Ig) | Ordered from The Jackson Laboratory, described in PMID:7926785 | Cat#:002595; MGI:2384162; RRID:IMSR_JAX:002595 | Transgenic mice with fully rearranged BCR |
| Cell line (*M. musculus*) | RAG2KO Pro-B cells | Kay Medina Lab | RRID:CVCL_B3QD | Cell line maintained by Medina lab See Materials and methods |
| Antibody | Anti-B220 BV785 (Rat monoclonal, RA3-6B2) | BioLegend | Cat#:103246; RRID:AB_2563256 | FC (1:200) |
| Antibody | Anti-B220 BV510 (Rat monoclonal, RA3-6B2) | BioLegend | Cat#:103248; RRID:AB_2650679 | FACS (1:200) |
| Antibody | Anti-Bcl2 PE (Mouse monoclonal, BCL/10C4) | BioLegend | Cat#:633508; RRID:AB_2290367 | FC (1:100) |
| Antibody | Anti-Bcl-xL AF488 (Rabbit monoclonal, 54H6) | Cell Signaling Technology | Cat#:2767S; RRID:AB_2274763 | FC (1:100) |
| Antibody | Anti-BP-1 biotin (Rat monoclonal, 6C3) | eBioscience | Cat#:12-5891-82; RRID:AB_466015 | FC (1:50) |
| Antibody | Anti-BP-1 BV605 (*Mus spretus* monoclonal, BP-1) | BD Biosciences | Cat#:745238; RRID:AB_2742824 | FC (1:50) |
| Antibody | Anti-BP-1 PE (Rat monoclonal, BP-1) | eBioscience | Cat#:12-5891-83; RRID:AB_466016 | FACS, FC (1:50) |
| Antibody | Anti-CD1d FITC (Rat monoclonal, 1B1) | BioLegend | Cat#:123508; RRID:AB_1236549 | FACS (1:100) |
| Antibody | Anti-CD2 FITC (Rat monoclonal, RM2-5) | eBioscience | Cat#:11-0021-81; RRID:AB_464872 | FC (1:100) |
| Antibody | Anti-CD4 biotin (Rat monoclonal, RM4-5) | BioLegend | Cat#:100508; RRID:AB_312711 | Negative selection (1:100) |
| Antibody | Anti-human CD5 APC (Mouse monoclonal, L17F12) | Tonbo Biosciences | Cat#:20-0058; RRID:AB_2621548 | FC [Cre reporter] (1:200) |
| Antibody | Anti-CD8$\alpha$ biotin (Rat monoclonal, 53-6.7) | BioLegend | Cat#:100704; RRID:AB_312743 | Negative selection (1:500) |
| Antibody | Anti-CD11b biotin (Rat monoclonal, M1/70) | BioLegend | Cat#:101204; RRID:AB_312787 | Negative selection (1:100) |
| Antibody | Anti-CD11c biotin (Armenian Hamster monoclonal, N418) | BioLegend | Cat#:117304; RRID:AB_313773 | Negative selection (1:100) |

*Appendix 1 Continued on next page*

*Appendix 1 Continued*

| Reagent type (species) or resource | Designation | Source or reference | Identifiers | Additional information |
|---|---|---|---|---|
| Antibody | Anti-CD19 BV510 (Rat monoclonal, 1D3) | BD Biosciences | Cat#:562956; RRID:AB_2737915 | FC (1:200) |
| Antibody | Anti-CD19 eFluor 450 (Rat monoclonal, eBio1D3) | eBioscience | Cat#:48-0193-82; RRID:AB_2734905 | FACS, FC (1:500) |
| Antibody | Anti-CD19 PE-Cy7 (Rat monoclonal, 6D5) | BioLegend | Cat#:115520; RRID:AB_313655 | FACS, FC (1:500) |
| Antibody | Anti-CD21/35 PerCP-Cy5.5 (Rat monoclonal, 7E9) | BioLegend | Cat#:123416; RRID:AB_1595490 | FACS (1:100) |
| Antibody | Anti-CD21/35 APC (Rat monoclonal, 7E9) | BioLegend | Cat#:123412; RRID:AB_2085160 | FC (1:200) |
| Antibody | Anti-CD23 APC (Rat monoclonal, B3B4) | BioLegend | Cat#:101614; RRID:AB_2103036 | FC (1:200) |
| Antibody | Anti-CD24 APC (Rat monoclonal, 30-F1) | BioLegend | Cat#:138506; RRID:AB_2565651 | FC (1:1000) |
| Antibody | Anti-CD24 FITC (Rat monoclonal, M1/69) | BioLegend | Cat#:101806; RRID:AB_312839 | FACS, FC (1:1000) |
| Antibody | Anti-CD25 PE (Rat monoclonal, PC61.5) | Tonbo Biosciences | Cat#:50-0251; RRID:AB_2621757 | FC (1:200) |
| Antibody | Anti-CD43 APC (Rat monoclonal, 1B11) | BioLegend | Cat#:121214; RRID:AB_528807 | FC (1:200) |
| Antibody | Anti-CD43 PerCP (Rat monoclonal, 1B11) | BioLegend | Cat#:121222; RRID:AB_893333 | FACS, FC (1:200) |
| Antibody | Anti-CD71 PE (Rat monoclonal, RI7217) | BioLegend | Cat#:113808; RRID:AB_313569 | FC (1:1000) |
| Antibody | Anti-CD93 PE (Rat monoclonal, AA4.1) | eBioscience | Cat#:12-5892-83; RRID:AB_466019 | FC (1:200) |
| Antibody | Anti-CD93 PE-Cy7 (Rat monoclonal, AA4.1) | BioLegend | Cat#:136506; RRID:AB_2044012 | FACS (1:100) |
| Antibody | Anti-CD127 PE-Cy7 (Rat monoclonal, A7R34) | BioLegend | Cat#:135014; RRID:AB_1937265 | FC (1:200) |
| Antibody | Anti-CDK2 PE (Rabbit monoclonal, 78B2) | Cell Signaling Technology | Cat#:14174; RRID:AB_2798413 | FC (1:50) |
| Antibody | Anti-Chk1 (Rabbit monoclonal, E250) | Abcam | Cat#:ab32531; RRID:AB_726821 | FC (1:200) |
| Antibody | Anti-E47/E2A FITC (Mouse monoclonal, G127-32) | BD Biosciences | Cat#:552510; RRID:AB_394408 | FC (1:500) |
| Antibody | Anti-EBF1 (Rabbit polyclonal) | Millipore Sigma | Cat#:ABE1294; RRID:AB_2893472 | FC (1:1000) |
| Antibody | Anti-FOXO1 PE (Rabbit monoclonal, C29H4) | Cell Signaling Technology | Cat#:14262S; RRID:AB_2798437 | FC (1:50) |
| Antibody | Anti-GR-1 biotin (Rat monoclonal, RB6-8C5) | BioLegend | Cat#:108404; RRID:AB_313369 | Negative selection (1:100) |
| Antibody | Anti-HO-1 FITC (Mouse monoclonal, HO-1–2) | Abcam | Cat#:ab69545; RRID:AB_2118659 | FC (1:50) |
| Antibody | Anti-IgA PE (Goat polyclonal IgG) | SouthernBiotech | Cat#:1040-09; RRID:AB_2794375 | FC (1:300) |
| Antibody | Anti-IgD PE (Rat monoclonal, 11–26c.2a) | BioLegend | Cat#:405706; RRID:AB_315028 | FC (1:200) |
| Antibody | Anti-IgG1 PE (Goat polyclonal F(ab')2 IgG) | SouthernBiotech | Cat#:1072-09; RRID:AB_2794434 | FC (1:1000) |

*Appendix 1 Continued on next page*

*Appendix 1 Continued*

| Reagent type (species) or resource | Designation | Source or reference | Identifiers | Additional information |
|---|---|---|---|---|
| Antibody | Anti-IgG2a PE (Goat polyclonal F(ab')2 IgG) | SouthernBiotech | Cat#:1082-09; RRID:AB_2794502 | FC (1:300) |
| Antibody | Anti-IgG2b PE (Goat polyclonal F(ab')2 IgG) | SouthernBiotech | Cat#:1092-09; RRID:AB_2794553 | FC (1:300) |
| Antibody | Anti-IgG3 PE (Goat polyclonal F(ab')2 IgG) | SouthernBiotech | Cat#:1102-09; RRID:AB_2784525 | FC (1:300) |
| Antibody | Anti-IgM APC-Cy7 (Rat monoclonal, RMM-1) | BioLegend | Cat#:406516; RRID:AB_10660305 | FC (1:100) |
| Antibody | Anti-IgM biotin (Rat monoclonal, RMM-1) | BioLegend | Cat#:406504; RRID:AB_315054 | Negative selection (1:100) |
| Antibody | Anti-IgM BV510 (Rat monoclonal, RMM-1) | BioLegend | Cat#:406531; RRID:AB_2650758 | FC (1:100) |
| Antibody | Anti-IgM FITC (Rat monoclonal, RMM-1) | BioLegend | Cat#:406506; RRID:AB_315056 | FC (1:100) |
| Antibody | Anti-IgM PE (Rat monoclonal, RMM-1) | BioLegend | Cat#:406508; RRID:AB_315058 | FACS (1:100) |
| Antibody | Anti-IgM PE-CF594 (Rat monoclonal, R6-60.2) | BD Biosciences | Cat#:562565; RRID:AB_2737658 | FACS (1:100) |
| Antibody | Anti-IKAROS AF488 (Rabbit monoclonal, D6N9Y) | Cell Signaling Technology | Cat#:89389S; RRID:AB_2800139 | FC (1:100) |
| Antibody | Anti-IRF4 PE (Rat monoclonal, 3E4) | eBioscience | Cat#:12-9858-82; RRID:AB_10852721 | FC (1:100) |
| Antibody | Anti-Ki-67 BV421 (Rat monoclonal, 16A8) | BioLegend | Cat#:652411; RRID:AB_2562663 | FC (1:200) |
| Antibody | Anti-Ki-67 PE (Rat monoclonal, 16A8) | BioLegend | Cat#:652404; RRID:AB_2561525 | FC (1:200) |
| Antibody | Anti-Mcl-1 PE (Rabbit monoclonal, D2W9E) | Cell Signaling Technology | Cat#:65617S; RRID:AB_2799688 | FC (1:50) |
| Antibody | Mouse IgG1 kappa Isotype FITC (Mouse monoclonal, P3.6.2.8.1) | eBioscience | Cat#:11-4714-81; RRID:AB_470021 | FC (concentration matched antibodies of interest) |
| Antibody | Mouse IgG1 kappa Isotype PE (Mouse monoclonal, MOPC-21) | BioLegend | Cat#:400111; RRID:AB_2847829 | FC (concentration matched antibodies of interest) |
| Antibody | Mouse IgG2b kappa Isotype FITC (Mouse monoclonal, 27-35) | BD Biosciences | Cat#:555742; RRID:AB_396085 | FC (concentration matched antibodies of interest) |
| Antibody | Anti-p53 AF488 (Mouse monoclonal, 1C12) | Cell Signaling Technology | Cat#:2015S; RRID:AB_2206297 | FC (1:50) |
| Antibody | Anti-PARP (Rabbit monoclonal, 46D11) | Cell Signaling Technology | Cat#:9532S; RRID:AB_659884 | FC (1:100) |
| Antibody | Anti-pATM (Ser1981) PE (Mouse monoclonal, 10H11.E12) | BioLegend | Cat#:651204; RRID:AB_2562655 | FC (1:500) |
| Antibody | Anti-PAX5 APC (Rat monoclonal, 1H9) | eBioscience | Cat#:17-9918-80; RRID:AB_10734230 | FC (1:200) |
| Antibody | Anti-pChk1 (Ser317) PE (Rabbit monoclonal, D12H3) | Cell Signaling Technology | Cat#:13959; RRID:AB_2893473 | FC (1:50) |
| Antibody | Anti-pH2A.X (Ser139) AF647 (Rabbit monoclonal, 20E3) | Cell Signaling Technology | Cat#:9720S; RRID:AB_10692910 | FC (1:100) |

*Appendix 1 Continued on next page*

| Reagent type (species) or resource | Designation | Source or reference | Identifiers | Additional information |
|---|---|---|---|---|
| Antibody | Rabbit IgG Isotype Control AF488 (Rabbit monoclonal) | Cell Signaling Technology | Cat#:4340; RRID:AB_561545 | FC (concentration matched antibodies of interest) |
| Antibody | Rabbit IgG Isotype Control AF647 (Rabbit monoclonal) | Cell Signaling Technology | Cat#:3452; RRID:AB_10695811 | FC (concentration matched antibodies of interest) |
| Antibody | Rabbit IgG Isotype Control PE (Rabbit monoclonal, DA1E) | Cell Signaling Technology | Cat#:5742; RRID:AB_10694219 | FC (concentration matched antibodies of interest) |
| Antibody | Rabbit IgG Monoclonal Isotype Control (Rabbit monoclonal, EPR25A) | Abcam | Cat#:ab172730; RRID:AB_2687931 | FC (concentration matched antibodies of interest) |
| Antibody | Rabbit IgG Polyclonal Isotype Control (Rabbit monoclonal) | Abcam | Cat#:ab171870; RRID:AB_2687657 | FC (concentration matched antibodies of interest) |
| Antibody | Rat IgG2a kappa Isotype APC (Rat monoclonal, RTK2758) | BioLegend | Cat#:400512; RRID:AB_2814702 | FC (concentration matched antibodies of interest) |
| Antibody | Rat IgG2a kappa Isotype PE (Rat monoclonal, RTK2758) | BioLegend | Cat#:400508; RRID:AB_326530 | FC (concentration matched antibodies of interest) |
| Antibody | Rat IgG2b kappa Isotype PE (Rat monoclonal, eB149/10H5) | eBioscience | Cat#:12-4031-82; RRID:AB_470042 | FC (concentration matched antibodies of interest) |
| Antibody | Anti-NK1.1 biotin (Mouse monoclonal, PK136) | BioLegend | Cat#:108704; RRID:AB_313391 | Negative selection (1:100) |
| Antibody | Anti-Ter119 biotin (Rat monoclonal, TER-119) | BioLegend | Cat#:116204; RRID:AB_313705 | Negative selection (1:100) |
| Antibody | Anti-TCRβ biotin (Armenian Hamster monoclonal, H57-597) | BioLegend | Cat#:109204; RRID:AB_313427 | Negative selection (1:100) |
| Antibody | Anti-TCRγδ biotin (Armenian Hamster monoclonal, eBioGL3) | eBioscience | Cat#:13-5711-82; RRID:AB_466668 | Negative selection (1:100) |
| Antibody | Anti-TdT PE (Mouse monoclonal, 19–3) | eBioscience | Cat#:12-5846-82; RRID:AB_1963620 | FC (1:200) |
| Antibody | Anti-VDAC1 (Rabbit polyclonal) | ProteinTech | Cat#:55259-1-AP; RRID:AB_10837225 | FC (1:100) |
| Sequence-based reagent | qPCR: ABCB7 probe | Thermo Fisher | AssayId:Mm01235269_m1 | FAM-MGB |
| Sequence-based reagent | qPCR: Eukaryotic 18S rRNA Endogenous Control | Thermo Fisher | Cat#:4352930 | FAM-MGB |
| Sequence-based reagent | qPCR: Rag1 probe | Thermo Fisher | AssayId:Mm01270936_m1 | FAM-MGB |
| Sequence-based reagent | qPCR: Rag2 probe | Thermo Fisher | AssayId:Mm00501300_m1 | FAM-MGB |
| Peptide, recombinant protein | Annexin V-FITC conjugate | BD Biosciences | Cat#:556420; AB_2665412 | FC (1:500) |
| Peptide, recombinant protein | Streptavidin BV510 | BioLegend | Cat#:405234 | FC (1:200) |
| Peptide, recombinant protein | Recombinant human BAFF | PeproTech | Cat#:310-13 | Class switch cultures |
| Peptide, recombinant protein | Recombinant murine IFN-γ | PeproTech | Cat#:315-05 | Class switch cultures |

| Reagent type (species) or resource | Designation | Source or reference | Identifiers | Additional information |
|---|---|---|---|---|
| Peptide, recombinant protein | Recombinant murine IL-4 | PeproTech | Cat#:214-14 | Class switch cultures |
| Peptide, recombinant protein | Recombinant murine IL-5 | PeproTech | Cat#:215-15 | Class switch cultures |
| Peptide, recombinant protein | Anti-δ dextran (mouse) | Fina Biosolutions | Cat#:FINABIO0001 | Class switch cultures |
| Peptide, recombinant protein | Recombinant human TGF-β1 | PeproTech | Cat#:100-21 | Class switch cultures |
| Commercial assay or kit | Click-iT Plus EdU AF488 Flow Cytometry Assay Kit | Thermo Fisher | Cat#:C10632 | EdU Assay |
| Commercial assay or kit | DNeasy Blood and Tissue Kit | Qiagen | Cat#:69504 | DNA purification |
| Commercial assay or kit | EasySep Mouse Streptavidin RapidSpheres Isolation Kit | STEMCELL Technologies | Cat#:19860A | Negative selection |
| Commercial assay or kit | MethoCult M3630 pre-B CFU kit | STEMCELL Technologies | Cat#:03630 | Pre-B CFU assay |
| Commercial assay or kit | OneTaq DNA polymerase | New England Biolabs | Cat#:M0480L | Semiquantitative PCR |
| Commercial assay or kit | RNeasy Mini Kit | Qiagen | Cat#:74104 | RNA isolation |
| Commercial assay or kit | SuperScript IV First-Strand Synthesis System | Thermo Fisher | Cat#:18091050 | cDNA synthesis |
| Chemical compound, drug | CFSE | Sigma | Cat#:21888 | Class switch cultures (2.5 µM) |
| Chemical compound, drug | FBS (qualified) | Thermo Fisher | Cat#:10437-028 | Class switch cultures |
| Chemical compound, drug | DAPI | BioLegend | Cat#:422801 | FC, cell cycle analysis (1:4000) |
| Chemical compound, drug | LPS | Sigma | Cat#:L-2630 | Class switch cultures |
| Software, algorithm | Illustrator CC 2019 | Adobe | RRID:SCR_010279 | Figure and image panel preparation |
| Software, algorithm | FlowJo v10.8 | BD Biosciences | RRID:SCR_008520 | FC analysis |
| Software, algorithm | GraphPad Prism v9 | GraphPad | RRID:SCR_002798 | Graphs and statistics |
| Other | BODIPY 581/591C11 C11 | Thermo Fisher | Cat#:D3861 | FC (2 µM) |
| Other | CellROX Green | Thermo Fisher | Cat#:C10444 | FC (5 µM) |
| Other | Ghost Dye Violet 450 (viability dye) | Tonbo Biosciences | Cat#:13-0863 | FACS, FC (1:1000) |
| Other | Ghost Dye Violet 510 (viability dye) | Tonbo Biosciences | Cat#:13-0870 | FACS, FC (1:1000) |
| Other | Ghost Dye Red 780 (viability dye) | Tonbo Biosciences | Cat#:13-0865 | FC (1:1000) |
| Other | MitoSOX Red | Thermo Fisher | Cat#:M36008 | FC (5 µM) |
| Other | MitoTracker Green FM | Thermo Fisher | Cat#:M7514 | FC (100 nM) |
| Other | Phen Green SK, diacetate | Thermo Fisher | Cat#:P14313 | FC (5 µM) |
| Other | Tetramethylrhodamine methyl ester perchlorate (TMRM) | Sigma | Cat#:T5428 | FC (100 nM) |

| Reagent type (species) or resource | Designation | Source or reference | Identifiers | Additional information |
|---|---|---|---|---|
| Other | ThiolTracker Violet | Thermo Fisher | Cat#:T10095 | FC (4 µM) |

## Appendix 2

| PCR primers | |
| --- | --- |
| **Semiquantitative PCR primers** | |
| **Designation** | **Sequence** |
| J$_H$3 primer | 5'- GTCTAGATTCTCACAAGAGTCCGATAGACCCTGG |
| V$_H$7183 primer | 5'-GCAGCTGGTGGAGTCTGG |
| V$_H$3609CDR2 primer | 5'-CAGGGCTGTGTTATA |
| V$_H$VGAM3.8CDR2 primer | 5'-CAAACCGTCCCTTGAA |
| V$_H$J558 primer | 5'-CAGGTCCAACTGCAGCAG |
| Actin_F primer | 5'-GGTGTCATGGTAGGTATGGGT |
| Actin_R primer | 5'-CGCACAATCTCACGTTCAG |

| qPCR primers | |
| --- | --- |
| **Designation** | **Sequence** |
| V$_H$7183GLT_F primer | 5'-CGGTACCAAGAASAMCCTGTWCCTGCAAATGASC |
| V$_H$7183GLT_R primer | 5'-GTCTCTCCGCGCCCCCTGCTGGTCC |
| V$_H$J558GLT_F primer | 5'-ACCATGGGATGGAGATGGATCTTTC |
| V$_H$J558GLT_F primer | 5'-CTCAGGATGTGGTTACAACACTGTG |
| qPCR: HPRT_F primer | 5'-AGGTTGCAAGCTTGCTGGT |
| HPRT_R primer | 5'-TGAAGTACTCATTATAGTCAAGGGCA |

