## [Decision Letter]

**Acceptance summary:**

This work focuses upon an important feature of biology, the interaction between basic cellular processes and cell differentiation. The study addresses how the ABCB7 transporter of iron impacts the differentiation of B lymphocytes.

**Decision letter after peer review:**

Thank you for submitting your article "ABCB7 is required for B cell development but not peripheral B cell homeostasis" for consideration by *eLife*. Your article has been reviewed by 3 peer reviewers, and the evaluation has been overseen by a Reviewing Editor and Betty Diamond as the Senior Editor. The following individual involved in review of your submission has agreed to reveal their identity: John Colgan (Reviewer #1).

The reviewers have discussed their reviews with one another, and the Reviewing Editor has drafted this to help you prepare a revised submission. Overall, the paper would be improved by greater information about the mechanisms by which ABCB7 impacts the processes examined.

Essential revisions:

1) Address question of how ABCB7 status impacts Ig H chain rearrangement, as some of the data presented support the role of ABCB7 in this process, while other data do not. The link between ABCB7 function and DNA damage/repair processes is not sufficiently strong. Additional experimentation, as suggested, could provide valuable insight.

2) Provide additional experimental support for the conclusion that ABCB7 is of lower importance in mature B cells compared to developing B cells. This is implied by data in CD23-Cre system, but not firmly established. Reviewers make several comments and suggestions that should be considered.

3) The conclusion that the major impact of ABCB7 occurs on the population C to C' developmental transition is not sufficiently convincing, based upon the results shown. Further clarification in data presentation and discussion would be very helpful to address this area of confusion.

4) There are quite a few technical issues raised by the reviewers. These need to be addressed. In some cases, this may just require clearer description and explanation; in others, additional experimentation will be needed.

*Reviewer #1:*

The manuscript by Lehrke et al., investigates the importance of the mitochondrial iron transporter ABCB7 for B cell development and mature B cell homeostasis. The principal methods used for this study are conditional deletion of the gene encoding ABCB7 in developing and mature B cells coupled to fairly sophisticated analysis of the effects of this using flow cytometry. The authors report that deletion of the gene for ABCB7 in early B cell progenitors results in a dramatic block in B cell development at a stage where antibody heavy chain gene rearrangement and cell proliferation must occur in order for B cell development to progress. This block occurs despite normal expression of transcription factors that are required for early B cell development. However, antibody heavy chain gene rearrangement is strongly impaired in the absence of ABCB7, suggesting that iron has a crucial role in this process, which requires DNA recombination and repair. Consistent with these results, transgenic expression of a rearranged heavy chain gene at least partially restores B cell development from ABCB7-deficient B cell progenitors. This result argues that an inability to successfully rearrange the antibody heavy chain gene and generate heavy chain protein is a main cause of the block in B cell development resulting from loss of ABCB7 expression. Some evidence is also provided to support the conclusion that DNA replication and thus cell proliferation is compromised in the absence of ABCB7 due to an inability to repair replication-induced DNA damage. In contrast to the dramatic effects on B cell progenitor development, deletion of the gene for ABCB7 in mature B cells has little effect, suggesting that ABCB7 is not essential for B cell homeostasis.

This manuscript is well organized and well written. The quality of the data shown is excellent, and most of the conclusions made are strong supported by the results presented. These are considered as strengths of the manuscript. Another strength is the characterization of how the loss of ABCB7 affects key events during B cell development; this thorough and methodical and reveals some important insights regarding the processes affected. A weakness of the manuscript is that the effects of ABCB7 deletion on mature B cell homeostasis is less well characterized; a few more experiments could extend this data in meaningful ways. Another weakness is that the effects on antibody heavy chain gene rearrangement and DNA replication were not teased apart.

The work presented by Lehrke et al., has impact on our understanding of the role of iron availability for progression of B cell development. It would appear that the appropriate handling of iron in B cell progenitors is critical for it to potentially serve as a cofactor for proteins that mediate antibody gene rearrangement and DNA replication and repair.

A key question that arises from this study is whether DNA replication in addition to immunoglobulin heavy chain gene rearrangement are affected by the absence of ABCB7. The authors provide evidence that both processes are affected; yet if DNA replication was strongly impaired, why would transgenic expression of a rearranged heavy chain gene largely suppress the effects of ABCB deficiency? Part of the answer could be that the lesion in DNA replication remains and may slow cell proliferation without halting development. To address this, the authors could look at EdU incorporation by pro-B cells and pre-B cells in WT and ABCB cKO mice that both carry the HEL-Ig transgene. If lesions in DNA replication persist, it would further support conclusion that DNA damage during replication is in part responsible for the phenotype.

The authors used CD23-cre to delete the gene encoding ABCB7 in mature B cells and didn't see any strong phenotype, suggesting ABCB7 is less important in these cells relative to developing B cells. Given a largely negative result was obtained, it would seem important to use PCR or Western blot analysis to confirm that the gene encoding ABCB7 is efficiently deleted in mature B cells by CD23-cre. Going further, if the authors wish to solidify the conclusion that ABCB7 is less important in mature B cells, they could stimulate WT and cKO cells in culture with LPS without or with the appropriate cytokines and measure proliferation and class-switching. Negative results here would support the idea that ABCB7 is largely dispensable in mature B cells; defects would support the conclusion that ABCB7 is required for DNA replication/proliferation and/or class switching by mature B cells.

*Reviewer #2:*

The report by Lehrke et al., describes a phenotypic analysis of B-lymphocyte development in mice with conditional deletion of the ABCB7 gene. Deletion in early B-cell progenitors using a mb1-cre driver result in a dramatic block in B-cell development. In contrast, deletion of the gene in mature cells, using a CD23-cre mouse strain, do not appear to cause observe any dramatic effects on either peripheral cell numbers or iron content in the cells. The developmental block induced by deletion of the ABCB7 gene in early progenitors could be overcome by expression of a BCR transgene, indicating a recombination deficiency as underlying cause of the developmental disruption. In consistency with this, the progenitor B-cells displayed an increased level of pH2gX. However, this was most prominent in proliferating cells that normally do not undergo VDJ recombination. Therefore, the authors suggest that the developmental block is a result of increased general DNA damage.

While the experimental designs in many ways are elegant, there are certain aspect of the analysis that complicates the interpretation of the data. Despite that the authors claim that the major block in differentiation occurs in the fraction C to C', a major part of the downstream analysis is made on a combined C/C' population. Hence, it is difficult to determine if the phenotypes observed is a consequence of a shift in populations or effects directly related to the function of ABCB7. It is also hard to understand how the expression of a functional B-cell receptor should be sufficient to overcome a general increase in DNA damage. Neither do the authors provide any direct link between DNA repair and ABCB7 function.

Hence, while the report clearly shows that ABCB7 is essential for normal early B-cell development, the mechanism by which mitochondrial iron-glutathione transport is linked to B-cell differentiation remains largely elusive.

1: The authors might want to investigate the deletion efficiencies in the more mature cell populations. As deletion of an essential gene cause a dramatic selection pressure this might result in that mature cells represent progenitors that has retained one or two functional alleles. This can complicate the interpretation of the data. Could this possibly be a reason for the apparent heterogeneity in the cKo cells in Figure 2J and 3A? As no phenotype is observed in Fig3S1, the interpretation of the data in Figure 3S1 would also be easier upon verification of functional ABCB7 deletion.

2: The authors could consider to be more stringent with their interpretations of their data. For instance, the reduced fraction B numbers in mice carrying a CD23 driver is considered as "largely unaffected" despite a p-value of 0.006. In contrast, the authors state " There was a trending, but not significant, decrease in HO-1 expression in Mb1-cre ABCB7 218 cKO pro-B cells (Figure 3B), confirming the previous findings that ABCB7 affects heme biosynthesis". indicating that a trend confirms a finding.

3: Much of the data analysis is obscured by the use of the combined C/C´ population. The authors claim that the C to C´ transition is targeted, hence in order to generate conclusive data, these populations should be analyzed separately. The data in figure 4S1B suffer from the same problem as this, as far as I can understand, is done on unsorted BM. In order to be informative sorted fraction B cells should have been seeded. The Y axis would benefit from inclusion of the number of seeded cells to be more informative.

4: I can be discussed if the term Wt can be used for a mix of different genotypes as done on this paper according to the MandM. This might explain the rather strange finding that the CD23 driver cause a significant reduction in fraction B cell numbers. This mixture of genotypes also generates rather strange statements such as the data are generated from 11 independent experiments (Figure legend 1C) despite that only 8 mice from each Ko is analyzed. It is also unclear what the data in the contour plots indicate, I guess one "representative" experiment. It is more stringent to use the average values with a std.

*Reviewer #3:*

In this manuscript by Lehrke et al., authors set out to describe the role of a poorly characterized cassette protein that regulates iron transport from mitochondria to cytosol. The lack of in depth scientific knowledge on how this molecule works makes it challenging for authors to develop their hypothesis. Nevertheless, authors elegantly show the developmental stage specific role of ABCB7 molecule in B cells. For the most part, experiments were designed and executed properly and controls are adequate. Authors show both positive and negative observations openly and their conclusion makes sense. However, the paper fails to explain how a redundant molecule that does not induce any pathological effects once knocked out (no apoptosis, no ferroptosis, no mitochondrial dysfunction, no ROS accumulation, no iron starvation) still manages to somehow affect a handful of genes only in a select subset of B cell precursors and create the demonstrated effects. It is not even clear whether the activities or biosynthesis of these genes/proteins that are differentially affected between knock out and wt are even influenced by iron homeostasis. Unbiased screening assays would have strengthened the manuscript by pointing out potential novel mechanisms or pathways.

Specific points:

General: Authors continuously use the word "trending" for statistically nonsignificant changes between groups. This is very confusing. If something is not significant, I do not see the point of highlighting that.

Figure 1: According to figure legend, number of repeats and number of mice used for each repeat is abnormally high. (eleven experiments with 8-17 mice per condition.) This makes hundreds of mice used. I do not see the rationale behind this many repeats considering that only handful of data points are actually shown in the figure and the findings are clear.

Figure 1 B: Showing absolute numbers is an ideal approach to support a hypothesis however, this strategy is impossible for bone marrow since unlike spleen which is a capsulated organ that can be meshed and counted as a whole bone marrow is not. So, flushing yields in bone marrow will depend on the amount of cutting on each end of the bone and the ability to capture the marrow without loss through needle flushing. While the data shown by authors is pretty significant and convincing, authors should also discuss that this method has limitations. Alternatively, authors can try to normalize by graphing each population as percentage of all viable isolated cells in their suspension.

Figure 1A-B: Authors build their manuscript on the observation of decreased Pro to Pre B (Fr C to C') cell in bone marrows of cKO mice. The contour plots on Figure 1A shows a slight change in percentages of Fraction C and C' however authors fail to highlight that point in the absolute cell number graph as both Fr. C and Fr. C' goes down in cell numbers. A blockage at a developmental stage is expected to increase the population before the blockage in which case this should be Fr.C. If authors see this as the selling point of the manuscript then they should show it more clearly. I suggest getting ratio of Fr. C over Fr. C' would help overcome the problems related to general hypocellularity in the bone marrow of cKO animals.

Figure 1 C: Gating of splenic B cell fractions should involve CD93 in order to discriminate the transitional cell pool which then can be further divided into T1-2-3 subpopulations using IgM and CD23. The gating strategy authors used is unconventional. If possible, it would be nice to see CD93 staining as well. Furthermore, using CD23 cre mouse will not be able to reveal the effects of ABCB7 KO in marginal zone and T1 cells which express little or no CD23. That limitation needs to be discussed in the text.

Figure 2 demonstrates convincing evidence on the effect of conditional KO in regulation of a select group of critical transcriptional mediators of pro to pre-B cell transition. While not necessary, an unbiased RNA seq experiment would have provided a far better portrait of the alterations in transcriptome level than qPCR experiments focusing on a handful of molecules.

Figure 3: The use of mitotracker green for general mitochondrial mass evaluation is not ideal. It has been shown multiple times that various factors including pH and mitochondrial dysfunction can affect the fluorescence intensity of mitotracker green. Authors should use fluorescent antibodies against specific mitochondrial markers such as VDAC-1, TOM-20 and COX-IV to comment on mitochondrial mass. Furthermore, TMRM staining alone does not give any idea about mitochondrial health. Changes in mitochondrial number between groups can easily cause shifts in TMRM intensity. Therefore, it needs to be normalized to exclude mitochondrial number in order to be used as a parameter that can detect mitochondrial performance. To do so, each sample needs to be divided into three groups. Groups need to be treated with either FCCP (basal staining indicator when mitochondria are depolarized) or oligomycin (maximum staining indicator when mitochondria are hyperpolarized) or nothing (unaltered current level) and TMRM values should be measured for each. Then using the formula 100x [ (MFI (untreated)- MFI(FCCP) )/ (MFI oligomycin)-MFI (FCCP)] percentage of maximum mitochondrial membrane potential used by each group can be found and this values are independent from the differences in mitochondrial numbers. I believe this may change the interpretation of results.

Figure 3: Authors should show under microscopy that Phen Green SK colocalizes with TMRM (or any other mitochondrial stain) in order to confirm the staining differences in Figure 1A are located to changes in mitochondrial levels. Occasionally, these dyes have nonspecific binding issues. This experiment will rule out that possibility.

Figure 3F: Mitochondrial ROS is not detected is a misleading claim (Page 13 lane268) I agree that the MitoSOX levels are identical between cKO and the control however, flow cytometry plots are relative for these dyes. Any mitochondria healthy or unhealthy produces some level of ROS and this is even thought to act as a part of normal physiology if the levels are within limits. So authors should amend this part.

Figure 4: It is a good idea to culture cells and measure Annexin levels which would rule out the possibility that apoptotic cells get cleared fast in bone marrow before they are detected. However, I think Annexin measurement at 16 h is not enough to make a statement. Authors should monitor viability using Live-Dead reagents at early and late time points (such as 3-6-9-12-16-24-48h or something like that) and graph changes. Additionally, commercially available apoptosis, necrosis kits that can detect early and late phases of apoptosis are available and can be beneficial to strengthen the statement.

Page 18, lane 377 onwards: It was not very clear how authors introduced the transgene to the mice for the reader. I figured out by looking at the methods section that this is actually crossbreeding the mice with MD4. However, it sounds from the text as if it was done through transfection. Authors should clearly state that they did cross breeding in this section to avoid any confusion. Also, the use of MD4 mice dates back to 1980s, I think the 1994 paper authors cited is not the original one. It should be a Goodnow paper as far as I remember, please double check.

Figure 7: Similar to Figure 2, authors hand pick a few markers to prove their point. While the selection is elegant and relevant, an unbiased approach would have been far more convincing. Furthermore, the list of markers tested are hard to follow and I did not understand why or how they get affected with slight change of iron levels? Are these molecules regulated by iron levels? How does iron relate to these changes? There is a gap in the story in here.

---

## [Author Response]

Essential revisions:1) Address question of how ABCB7 status impacts Ig H chain rearrangement, as some of the data presented support the role of ABCB7 in this process, while other data do not. The link between ABCB7 function and DNA damage/repair processes is not sufficiently strong. Additional experimentation, as suggested, could provide valuable insight.

We have performed additional experiments as requested by the reviewers. Please see our specific answers in the point-by-point response below. In brief, ABCB7-deficient pro-B cells have decreased heavy chain rearrangement, μHC expression, decreased proliferation, and increased DNA damage in proliferating cells. B cell development is rescued in Mb1-cre ABCB7 cKO mice when interbred with the MD4 HEL-Ig transgene. We found that proliferation was similar in B220^+^CD19^+^CD43^+^CD127^+^ pro-B cells from Hel-Ig transgenic and HEL-Ig transgenic Mb1-cre ABCB7 cKO mice, although increased DNA damage persisted in the absence of ABCB7 (new Figure 9). Thus, the Hel-Ig transgene restores both Ig expression and proliferation. We believe this may be due to the differences in signals received by B220^+^CD19^+^CD43^+^ pro-B cells in these strains of mice. While normally B220^+^CD19^+^CD43^+^ pro-B cells receive signals through the pre-BCR, in HEL-Ig transgenic mice B220^+^CD19^+^CD43^+^ pro-B cells express high levels of IgM and IgD on the cell surface. Analysis of proliferation of ABCB7-deficient splenic B cells under class switching conditions demonstrated that the severity of the defect depended upon the stimuli utilized (new Figure 8).

As to the cause for increased DNA damage in the absence of ABCB7, there is a direct link between ABCB7 and DNA repair, as the iron-sulfur clusters exported from the mitochondria by ABCB7 are cofactors in numerous DNA repair enzymes including DNA primase, all replicative DNA polymerases, the helicases Dna2, FancJ, and XPD, and the glycosylases Endo III and MutY ^1–9^. Therefore, a disruption in iron-sulfur cluster levels may impair the function of one or numerous DNA repair enzymes in the absence of ABCB7.

2) Provide additional experimental support for the conclusion that ABCB7 is of lower importance in mature B cells compared to developing B cells. This is implied by data in CD23-Cre system, but not firmly established. Reviewers make several comments and suggestions that should be considered (e.g. Rev 1, ¶2; Rev 2, Points 1 and 2; Rev 3, points re Figure 1).

We thank the reviewers for this important suggestion. We have added qPCR data demonstrating efficient deletion of ABCB7 in follicular and marginal zone B cells from CD23-cre ABCB7 cKO mice. In new Figure 8, we now show that ABCB7 is important for splenic B cell proliferation and class switching after stimulation. These data demonstrate that ABCB7 is not required for peripheral B cell homeostasis, as the numbers and proportions of peripheral B cells were unchanged in CD23-cre ABCB7 cKO mice (Figure 1), but is important for proliferation and class switching.

3) The conclusion that the major impact of ABCB7 occurs on the population C to C' developmental transition is not sufficiently convincing, based upon the results shown. Further clarification in data presentation and discussion would be very helpful to address this area of confusion.

We have adjusted the text to clarify that we believe the block in B cell development in the bone marrow is occurring in pro-B cells leading to a severe defect in B cell development at subsequent stages, rather than specifically at the Fr. C to Fr. C’ transition. Please see specific responses to the reviewer comments below.

4) There are quite a few technical issues raised by the reviewers. These need to be addressed. In some cases, this may just require clearer description and explanation; in others, additional experimentation will be needed.

These have been addressed in our point-by-point responses to each reviewer below.

Reviewer #1:A key question that arises from this study is whether DNA replication in addition to immunoglobulin heavy chain gene rearrangement are affected by the absence of ABCB7. The authors provide evidence that both processes are affected; yet if DNA replication was strongly impaired, why would transgenic expression of a rearranged heavy chain gene largely suppress the effects of ABCB deficiency? Part of the answer could be that the lesion in DNA replication remains and may slow cell proliferation without halting development. To address this, the authors could look at EdU incorporation by pro-B cells and pre-B cells in WT and ABCB cKO mice that both carry the HEL-Ig transgene. If lesions in DNA replication persist, it would further support conclusion that DNA damage during replication is in part responsible for the phenotype.

We thank the reviewer for these suggestions and have added these results in new Figure 9. Pro-B cells (defined by B220^+^CD19^+^CD43^+^CD127^+^) from Hel-Ig Mb1-cre ABCB7 cKO and HEL-Ig WT mice had equivalent proportions of EdU incorporation and similar cell cycle status as analyzed by DAPI. This is in contrast to Mb1-cre ABCB7 pro-B cells which had a reduced proportion of cells that incorporated EdU (Figure 7). Interestingly, similar to Mb1-cre ABCB7 cKO pro-B cells, HEL-Ig Mb1-cre ABCB7 cKO pro-B cells had greater pH2A.X expression in proliferating cells and reduced EdU gMFI, suggesting a slower rate of DNA replication and increased DNA damage as compared to Hel-Ig WT pro-B cells. Unlike ABCB7-deficient pro-B cells, HEL-Ig Mb1-cre ABCB7 cKO pro-B cells had normal Ki-67 expression compared to HEL-Ig WT pro-B cells. This suggests that the presence of the fully rearranged MD4 HEL-Ig transgenic BCR is able to rescue proliferation in the absence of ABCB7, despite the presence of DNA damage. We believe this may be due to differences in the signals received by pro-B cells in these different models. In HEL-Ig transgenic mice, fully rearranged IgM and IgD receptors are expressed on B220^+^CD19^+^CD43^+^ pro-B cells at high levels; while in non-transgenic mice, pro-B cells signal through the pre-BCR, which is expressed at low levels. Therefore, we believe differences in BCR vs pre-BCR signaling may be responsible for the restored proliferation in HEL-Ig Mb1-cre ABCB7 cKO pro-B cells. In support of this, we observed that the extent of proliferation of ABCB7-deficient splenic B cells was dependent upon the stimulation conditions (new Figure 8).

The authors used CD23-cre to delete the gene encoding ABCB7 in mature B cells and didn't see any strong phenotype, suggesting ABCB7 is less important in these cells relative to developing B cells. Given a largely negative result was obtained, it would seem important to use PCR or Western blot analysis to confirm that the gene encoding ABCB7 is efficiently deleted in mature B cells by CD23-cre.

We performed qPCR analysis on FACS sorted follicular (CD19^+^AA4.1^+^CD21/35^+^IgM^+^) and marginal zone (CD19^+^AA4.1^+^CD21/35^hi^IgM^hi^CD1d^+^) B cells from CD23-cre ABCB7 cKO mice. As expected, we found that both populations were deficient for *Abcb7* expression (new Figure 1 —figure supplement 2C).

Going further, if the authors wish to solidify the conclusion that ABCB7 is less important in mature B cells, they could stimulate WT and cKO cells in culture with LPS without or with the appropriate cytokines and measure proliferation and class-switching. Negative results here would support the idea that ABCB7 is largely dispensable in mature B cells; defects would support the conclusion that ABCB7 is required for DNA replication/proliferation and/or class switching by mature B cells.

We thank the reviewer for their suggestion to stimulate peripheral B cells from CD23-cre ABCB7 cKO mice. We performed an analysis of proliferation and class switching in splenic B cells from CD23-cre ABCB7 cKO and littermate controls. Enriched CD19^+^ cells from these mice were stimulated with LPS and various cytokines and/or anti-δ-dextran to induce activation, proliferation, and class switching to IgG1, IgG2a, IgG2b, IgG3, and IgA. Despite normal peripheral B cell homeostasis, ABCB7-deficient peripheral B cells had a striking defect in proliferation and class switching (new Figure 8). Interestingly, the severity of the defect in proliferation was highly dependent upon the stimulation conditions used, demonstrating that the type of stimuli received has differential effects on the extent of proliferation. Furthermore, ABCB7-deficient cells in these cultures had a larger proportion of undivided cells that lacked Ki-67 expression, suggesting these cells have lost proliferation potential (new Figure 8 —figure supplement 3). We hypothesize that the defect in class switching is largely a result of impaired proliferation, which has been described thoroughly in the literature ^10–14^. No differences were observed in the expression of class switched antibody isotypes in the splenic B cells from CD23-cre ABCB7 cKO mice and littermate controls (new Figure 8 —figure supplement 1). Thus, while ABCB7 deficiency is dispensable for maintenance of splenic B cell numbers at homeostasis, ABCB7 is required for splenic B cell proliferation.

Reviewer #2:The report by Lehrke et al., describes a phenotypic analysis of B-lymphocyte development in mice with conditional deletion of the ABCB7 gene. Deletion in early B-cell progenitors using a mb1-cre driver result in a dramatic block in B-cell development. In contrast, deletion of the gene in mature cells, using a CD23-cre mouse strain, do not appear to cause observe any dramatic effects on either peripheral cell numbers or iron content in the cells. The developmental block induced by deletion of the ABCB7 gene in early progenitors could be overcome by expression of a BCR transgene, indicating a recombination deficiency as underlying cause of the developmental disruption. In consistency with this, the progenitor B-cells displayed an increased level of pH2gX. However, this was most prominent in proliferating cells that normally do not undergo VDJ recombination. Therefore, the authors suggest that the developmental block is a result of increased general DNA damage.While the experimental designs in many ways are elegant, there are certain aspect of the analysis that complicates the interpretation of the data. Despite that the authors claim that the major block in differentiation occurs in the fraction C to C', a major part of the downstream analysis is made on a combined C/C' population. Hence, it is difficult to determine if the phenotypes observed is a consequence of a shift in populations or effects directly related to the function of ABCB7. It is also hard to understand how the expression of a functional B-cell receptor should be sufficient to overcome a general increase in DNA damage. Neither do the authors provide any direct link between DNA repair and ABCB7 function.Hence, while the report clearly shows that ABCB7 is essential for normal early B-cell development, the mechanism by which mitochondrial iron-glutathione transport is linked to B-cell differentiation remains largely elusive.1: The authors might want to investigate the deletion efficiencies in the more mature cell populations. As deletion of an essential gene cause a dramatic selection pressure this might result in that mature cells represent progenitors that has retained one or two functional alleles. This can complicate the interpretation of the data. Could this possibly be a reason for the apparent heterogeneity in the cKo cells in Figure 2J and 3A? As no phenotype is observed in Fig3S1, the interpretation of the data in Figure 3S1 would also be easier upon verification of functional ABCB7 deletion.

As stated above, qPCR was performed and ABCB7 is efficiently deleted in follicular and marginal zone B cells from CD23-cre ABCB7 cKO mice. Thus, the lack of Phen Green quenching, indicative of iron overload, in splenic B cells in Figure 3 —figure supplement 1 is not due to lack of ABCB7 deletion in that model. While there is heterogeneity observed with Phen Green quenching in ABCB7-deficient pro-B cells, we believe that this simply indicates that time is needed to accumulate iron.

2: The authors could consider to be more stringent with their interpretations of their data. For instance, the reduced fraction B numbers in mice carrying a CD23 driver is considered as "largely unaffected" despite a p-value of 0.006. In contrast, the authors state " There was a trending, but not significant, decrease in HO-1 expression in Mb1-cre ABCB7cKO pro-B cells (Figure 3B), confirming the previous findings that ABCB7 affects heme biosynthesis". indicating that a trend confirms a finding.

We have eliminated any discussion of trends and limited interpretation to observed changes with statistical significance of p < 0.05.

As stated by Reviewer 3, it is impossible to completely isolate all the cells from the bone marrow. Based on their suggestions, we have analyzed the frequency of each developing B cell population in Figure 1. Based on frequency, there is no statistical difference in the frequency of Fr. B cells from Cd23-cre ABCB7 cKO mice as compared to controls. The analysis of absolute numbers of bone marrow cells has been moved to Figure 1 —figure supplement 1.

3: Much of the data analysis is obscured by the use of the combined C/C´ population. The authors claim that the C to C´ transition is targeted, hence in order to generate conclusive data, these populations should be analyzed separately. The data in figure 4S1B suffer from the same problem as this, as far as I can understand, is done on unsorted BM. In order to be informative sorted fraction B cells should have been seeded. The Y axis would benefit from inclusion of the number of seeded cells to be more informative.

We agree and have clarified the text and figures to demonstrate that we believe that the better way to describe the B cell phenotype is that there is a defect initiating at the pro-B cell stage, leading to a severe loss of subsequent development stages. Regarding the pre-B CFU assay, we followed the manufacturer’s instructions to seed total bone marrow. This kit is designed to generate pre-B CFU, and sorting Fr. B cells would not have worked in this assay. In addition to IL-7 for robust expansion, Fr. B cells also require stromal cell support as well as Flt3 ligand and SCF signals, which are not provided by the pre-B CFU assay. To clarify the data, we have moved the pre-B CFU data Figure 2 —figure supplement 1 as it better supports failure to generate pre-B cells. We have also added the seeded cell density to the figure and figure legend.

4: I can be discussed if the term Wt can be used for a mix of different genotypes as done on this paper according to the MandM. This might explain the rather strange finding that the CD23 driver cause a significant reduction in fraction B cell numbers. This mixture of genotypes also generates rather strange statements such as the data are generated from 11 independent experiments (Figure legend 1C) despite that only 8 mice from each Ko is analyzed. It is also unclear what the data in the contour plots indicate, I guess one "representative" experiment. It is more stringent to use the average values with a std.

We thank the reviewer for their comments and concern. As stated above, analysis of Hardy fractions as a percentage of live cells revealed that there was not a difference in the proportion of different Hardy fractions in the CD23-cre ABCB7 cKO mice.

We apologize for the confusion regarding the number of independent experiments and analyzed mice figure legends. Figure legends have been updated to clarify that the number of mice reported is the total number of mice analyzed across all independent experiments. Additionally, we have reanalyzed splenic B cell populations in Figure 1 at the suggestion of Reviewer #3. Therefore, the figure legend now reads “…contour plots are representative of seven independent experiments (total of 7-12 mice/group).” This indicates that seven independent experiments were performed, and over the course of these experiments a total of 7-12 mice/group were analyzed (e.g., 12 WT mice, 7 Mb1-cre ABCB7 cKO mice, and 8 CD23-cre ABCB7 cKO mice).

Contour plots are described as, “representative of n independent experiments.” By this we mean that contour plot data shown in a figure is visually equivalent to other independent replicates of that experiment. Throughout our manuscript, we show representative flow cytometry data and then quantify across experiments in a subsequent panel for rigor and reproducibility.

Reviewer #3:Specific points:General: Authors continuously use the word "trending" for statistically nonsignificant changes between groups. This is very confusing. If something is not significant, I do not see the point of highlighting that.

We have eliminated any discussion of trends and limited interpretation to observed changes with statistical significance of p < 0.05.

Figure 1: According to figure legend, number of repeats and number of mice used for each repeat is abnormally high. (eleven experiments with 8-17 mice per condition.) This makes hundreds of mice used. I do not see the rationale behind this many repeats considering that only handful of data points are actually shown in the figure and the findings are clear.

We apologize for the confusion regarding the number of independent experiments and analyzed mice figure legends. Figure legends have been updated to clarify that the number of mice reported is the total number of mice analyzed across all independent experiments. For example, a total of 6-11 mice/group were analyzed for Figure 1A (e.g., 11 WT mice, 7 Mb1-cre ABCB7 cKO mice, and 6 CD23-cre ABCB7 cKO mice). Unless otherwise stated in the figure legend, each data point on a bar graph represents one mouse. Please note that splenic B cell populations presented in Figure 1 have been updated to include the use of CD93/AA4.1 and the number of experiments and mice have been updated.

Figure 1 B: Showing absolute numbers is an ideal approach to support a hypothesis however, this strategy is impossible for bone marrow since unlike spleen which is a capsulated organ that can be meshed and counted as a whole bone marrow is not. So, flushing yields in bone marrow will depend on the amount of cutting on each end of the bone and the ability to capture the marrow without loss through needle flushing. While the data shown by authors is pretty significant and convincing, authors should also discuss that this method has limitations. Alternatively, authors can try to normalize by graphing each population as percentage of all viable isolated cells in their suspension.

We agree that any bone marrow harvest procedure has a limitation in that it is impossible to completely flush/remove all bone marrow cells, as cells may become trapped in the bones. To clarify, bone marrow was harvested by centrifugation^15^ rather than needle flushing. We have found the centrifugation method provides a more thorough bone marrow harvest compared to needle flushing or crushing methods.

We have analyzed the Figure 1 bone marrow data by graphing each Hardy fraction population as a percentage of all viable cells isolated. Using cell frequency, there continues to be a striking difference in B cell development in Mb1-cre ABCB7 cKO mice. This analysis also demonstrated that the proportions of Hardy fractions in CD23-cre ABCB7 cKO mice were not statistically different.

Figure 1A-B: Authors build their manuscript on the observation of decreased Pro to Pre B (Fr C to C') cell in bone marrows of cKO mice. The contour plots on Figure 1A shows a slight change in percentages of Fraction C and C' however authors fail to highlight that point in the absolute cell number graph as both Fr. C and Fr. C' goes down in cell numbers. A blockage at a developmental stage is expected to increase the population before the blockage in which case this should be Fr.C. If authors see this as the selling point of the manuscript then they should show it more clearly. I suggest getting ratio of Fr. C over Fr. C' would help overcome the problems related to general hypocellularity in the bone marrow of cKO animals.

We agree and have altered the manuscript to state that the defect is in pro-B cells leading to a severe loss of pre-B cells, and subsequent stages. Because we observed a statistically significant decrease in Fr. B and Fr. C cell numbers in Mb1-cre ABCB7 cKO mice, we have clarified in the text that we believe the block in B cell development in these mice is generally occurring at the pro-B cell stage. We have also added the ratio of Fr. C and Fr. C’ cells. We found that compared to WT mice, Mb1-cre ABCB7 cKO mice had a significantly higher ratio of Fr. C cells over Fr. C’ cells (new Figure 1 —figure supplement 1B).

Figure 1 C: Gating of splenic B cell fractions should involve CD93 in order to discriminate the transitional cell pool which then can be further divided into T1-2-3 subpopulations using IgM and CD23. The gating strategy authors used is unconventional. If possible, it would be nice to see CD93 staining as well. Furthermore, using CD23 cre mouse will not be able to reveal the effects of ABCB7 KO in marginal zone and T1 cells which express little or no CD23. That limitation needs to be discussed in the text.

We thank the reviewer for their suggestion and comments. We have updated Figure 1 to include CD93 (AA4.1) and a different gating strategy for splenic B cells. We now analyze the following CD19^+^ populations: T1 (AA4.1^+^CD21/35-IgM^+^CD23^-^), T2 (AA4.1^+^CD21/35^-^IgM^+^CD23^+^), T3 (AA4.1^+^CD21/35^+^IgM^+^), FO (AA4.1^-^CD21/35^+^IgM^+^), and MZ (AA4.1-CD21/35^hi^IgM^hi^).

Regarding CD23-cre expression in T1 B cells, we agree that these mice express little CD23 and therefore would not have robust Cre expression. As described above, we analyzed expression of a human CD5 reporter that is linked to CD23-cre expression via an IRES. We did not observe robust huCD5 reporter expression in T1 B cells, as expected. The reporter was expressed in T2, T3, and FO B cells. As described above, we performed qPCR analysis for *Abcb7* expression in FACS sorted follicular and marginal zone B and found that both populations efficiently deleted *Abcb7*. We have added discussion of the limitation to examining T1 cells as suggested.

Figure 2 demonstrates convincing evidence on the effect of conditional KO in regulation of a select group of critical transcriptional mediators of pro to pre-B cell transition. While not necessary, an unbiased RNA seq experiment would have provided a far better portrait of the alterations in transcriptome level than qPCR experiments focusing on a handful of molecules.

We thank the reviewer for their suggestion, and plan to do this in the future.

Figure 3: The use of mitotracker green for general mitochondrial mass evaluation is not ideal. It has been shown multiple times that various factors including pH and mitochondrial dysfunction can affect the fluorescence intensity of mitotracker green. Authors should use fluorescent antibodies against specific mitochondrial markers such as VDAC-1, TOM-20 and COX-IV to comment on mitochondrial mass. Furthermore, TMRM staining alone does not give any idea about mitochondrial health. Changes in mitochondrial number between groups can easily cause shifts in TMRM intensity. Therefore, it needs to be normalized to exclude mitochondrial number in order to be used as a parameter that can detect mitochondrial performance. To do so, each sample needs to be divided into three groups. Groups need to be treated with either FCCP (basal staining indicator when mitochondria are depolarized) or oligomycin (maximum staining indicator when mitochondria are hyperpolarized) or nothing (unaltered current level) and TMRM values should be measured for each. Then using the formula 100x [ (MFI (untreated)- MFI(FCCP) )/ (MFI oligomycin)-MFI (FCCP)] percentage of maximum mitochondrial membrane potential used by each group can be found and this values are independent from the differences in mitochondrial numbers. I believe this may change the interpretation of results.

We have added flow cytometry analysis of VDAC1 expression and found similar expression in ABCB7-deficient pro-B cells as compared to WT pro-B cells (new panel in Figure 3). Thus, there is similar mitochondrial mass by either MitoTracker Green and VDAC1 expression, providing support that the lack of difference in TMRM staining is not due to alterations in mitochondrial mass.

Figure 3: Authors should show under microscopy that Phen Green SK colocalizes with TMRM (or any other mitochondrial stain) in order to confirm the staining differences in Figure 1A are located to changes in mitochondrial levels. Occasionally, these dyes have nonspecific binding issues. This experiment will rule out that possibility.

Unlike other cellular dyes that analyze fluorescence intensity, Phen Green is read out as fluorescence quenching. This is because Phen Green is a general cell dye that brightly stains the both the cytosol and organelles. This fluorescence is partially quenched in the presence of heavy metal ions, but not completely absent. Therefore, it would be difficult to assess the localization of fluorescence quenching to the mitochondria, especially in developing lymphocytes which do not have an abundance of cytoplasm.

Figure 3F: Mitochondrial ROS is not detected is a misleading claim (Page 13 lane268) I agree that the MitoSOX levels are identical between cKO and the control however, flow cytometry plots are relative for these dyes. Any mitochondria healthy or unhealthy produces some level of ROS and this is even thought to act as a part of normal physiology if the levels are within limits. So authors should amend this part.

We thank the author for the suggestion and have amended the text.

Figure 4: It is a good idea to culture cells and measure Annexin levels which would rule out the possibility that apoptotic cells get cleared fast in bone marrow before they are detected. However, I think Annexin measurement at 16 h is not enough to make a statement. Authors should monitor viability using Live-Dead reagents at early and late time points (such as 3-6-9-12-16-24-48h or something like that) and graph changes. Additionally, commercially available apoptosis, necrosis kits that can detect early and late phases of apoptosis are available and can be beneficial to strengthen the statement.

In Figure 8 —figure supplement 2, we have examined viability over four days in culture of magnetically enriched splenic B cells from WT and CD23-cre ABCB7 cKO mice. No statistical differences were observed in viability of unstimulated cells across the time course.

Page 18, lane 377 onwards: It was not very clear how authors introduced the transgene to the mice for the reader. I figured out by looking at the methods section that this is actually crossbreeding the mice with MD4. However, it sounds from the text as if it was done through transfection. Authors should clearly state that they did cross breeding in this section to avoid any confusion. Also, the use of MD4 mice dates back to 1980s, I think the 1994 paper authors cited is not the original one. It should be a Goodnow paper as far as I remember, please double check.

We apologize for the confusion regarding the use MD4 HEL-Ig mice. We have altered the text to clarify their use. Additionally, the reviewer was correct regarding the 1988 Goodnow paper and the citation has been changed.

Figure 7: Similar to Figure 2, authors hand pick a few markers to prove their point. While the selection is elegant and relevant, an unbiased approach would have been far more convincing. Furthermore, the list of markers tested are hard to follow and I did not understand why or how they get affected with slight change of iron levels? Are these molecules regulated by iron levels? How does iron relate to these changes? There is a gap in the story in here.

We thank the reviewer for their comments. We plan to perform RNA-seq in the future for an unbiased examination of gene expression changes.